# PTEN differentially regulates endocytosis, migration, and proliferation in the enteric protozoan parasite *Entamoeba histolytica*

**Samia Kadri**[1☯], **Kumiko Nakada-Tsukui**[2☯], **Natsuki Watanabe**[1], **Ghulam Jeelani**[1], **Tomoyoshi Nozaki**[1]*

**1** Department of Biomedical Chemistry, Graduate School of Medicine, The University of Tokyo, Tokyo, Japan, **2** Department of Parasitology, National Institute of Infectious Diseases, Tokyo, Japan

☯ These authors contributed equally to this work.
* nozaki@m.u-tokyo.ac.jp

**Data Availability Statement:** All relevant data are within the manuscript and its Supporting Information files.

## Abstract

PTEN is a lipid phosphatase that is highly conserved and involved in a broad range of biological processes including cytoskeletal reorganization, endocytosis, signal transduction, and cell migration in all eukaryotes. Although regulation of phosphatidylinositol (3,4,5)-trisphosphate [PtdIns(3,4,5)P$_3$] signaling via PTEN has been well established in model organisms and mammals, it remains elusive in the parasitic protist *E. histolytica*, which heavily relies on PtdIns phosphate(s)-dependent membrane traffic, migration, and phago- and trogocytosis for its pathogenesis. In this study, we characterized the major PTEN from *E. histolytica*, EhPTEN1, which shows the highest expression at the transcript level in the trophozoite stage among 6 possible PTENs, to understand the significance of PtdIns(3,4,5)P$_3$ signaling in this parasite. Live imaging of GFP-EhPTEN1 expressing amebic trophozoites showed localization mainly in the cytosol with a higher concentration at pseudopods and the extending edge of the phago- and trogocytic cups. Furthermore, quantitative analysis of phago- and trogocytosis using a confocal image cytometer showed that overexpression of EhPTEN1 caused reduction in trogo- and phagocytosis while transcriptional gene silencing of *EhPTEN1* gene caused opposite phenotypes. These data suggest that EhPTEN1 has an inhibitory role in these biological processes. Conversely, EhPTEN1 acts as a positive regulator for fluid-phase and receptor-mediated endocytosis in *E. histolytica* trophozoites. Moreover, we showed that EhPTEN1 was required for optimal growth and migration of this parasite. Finally, the phosphatase activity of EhPTEN1 towards PtdIns(3,4,5)P$_3$ was demonstrated, suggesting that the biological roles of EhPTEN1 are likely linked to its catalytic function. Taken together, these results indicate that EhPTEN1 differentially regulates multiple cellular activities essential for proliferation and pathogenesis of the organism, via PtdIns(3,4,5)P$_3$ signaling. Elucidation of biological roles of PTEN and PtdIns(3,4,5)P$_3$ signaling at the molecular levels promotes our understanding of the pathogenesis of this parasite.

**Funding:** This work was supported by Core-to-Core Program, (JPJSCCB20190010) from the Japan Society for the Promotion of Science, Grants-in-Aid for Scientific Research (B) (JP18H02650 and JP21H02723 to TN) from the Japan Society for the Promotion of Science, Grant for Science and Technology Research Partnership for Sustainable Development (SATREPS) from AMED and Japan International Cooperation Agency (JICA) (JP20jm0110022) to TN, Grant for research on emerging and re-emerging infectious diseases from Japan Agency for Medical Research and Development (AMED, JP20fk0108138 to TN) and (AMED, JP20fk0108139 to KNT), Grants-in-Aid for Scientific Research (B) and Scientific Research on Innovative Areas (JP19H03463 and JP20H05353 to KNT) from Ministry of Education, Culture, Sports, Science and Technology (MEXT) or Japan Society for Promotion of Sciences (JSPS), Grant-in-Aid for Research Activity start-up (JP20K22758 to NW), Sasagawa Scientific Research Grant from The Japan Science Society (2020-4044 to NW), and Grant-in-Aid for Young Scientists from Wakate (JP21K15426 to NW). The funders had no role in study design, data collection and analysis, decision to publish, or preparation of the manuscript.

**Competing interests:** The authors have declared that no conflict interests exist.

## Author summary

*Entamoeba histolytica* is an intestinal protozoan parasite that causes amoebic dysentery and liver abscesses in humans. It has been well understood how the amoeba's ability to ingests and destroy human cells and invade tissues contributes to disease symptoms such as bloody diarrhea. The underlying mechanisms for such activities, called pathogenicity, include trafficking (transport) and secretion of cytolytic proteins, migration (ameboid movement), and ingestion and destruction of human cells, heavily rely on the signal transduction system via metabolism (synthesis and decomposition) of phosphoinositides (phosphatidylinositols containing 0–3 phosphates), and downstream regulation of cytoskeleton (dynamic network of interlinking protein filaments, such as actin, in the cytoplasm). In this study, we characterized one enzyme called EhPTEN1, which degrades and inactivate $PtdIns(3,4,5)P_3$. We have shown that EhPTEN1 is involved in migration, internalization of soluble and solid materials (endocytosis, trogo-, and phagocytosis). EhPTEN1 apparently regulates cell migration, endocytosis, trogo-, phagocytosis, and proliferation in a complex fashion. Our findings help in the elucidation of the physiological significance of PTEN and cellular events regulated via phosphoinositides in this enteric parasite and other pathogenic parasites, and may potentially lead to the development of new control measures against parasitic diseases.

## Introduction

Phosphatidylinositol phosphates (PIPs) are membrane phospholipids that play pivotal roles in a variety of biological processes such as cytoskeletal reorganization, vesicular trafficking, endocytosis, signal transduction, ion channel activation, and cell migration [1,2]. There are seven different species of PIPs in mammalian cells including three phosphatidylinositol monophosphate, three phosphatidylinositol biphosphate, and one phosphatidylinositol triphosphate [2]. PIPs kinases and phosphatases regulate the cellular function of PIPs through reversible phosphorylation and de-phosphorylation [3]. PTEN (phosphatase and tensin homologue) is a lipid phosphatase that dephosphorylates phosphatidylinositol (3,4,5)-trisphosphate [$PtdIns(3,4,5)P_3$] to phosphatidylinositol (4,5)-bisphosphate [$PtdIns(4,5)P_2$], thus depleting cellular signaling processes downstream of $PtdIns(3,4,5)P_3$ [4]. $PtdIns(3,4,5)P_3$ acts as a secondary messenger which activates the proto-oncogenic PI3K–AKT signaling pathway [5]. PTEN plays a crucial role in cell proliferation through its cytoplasmic phosphatase activity against the PI3K–AKT cascade [6]. Also, PTEN regulates cell polarity and migration via the establishment of a $PtdIns(3,4,5)P_3$-$PtdIns(4,5)P_2$ gradient [7,8]. Many human cancers are associated to PTEN mutations, including endometrial tumors, glioblastoma, prostate carcinoma, melanoma, and hereditary cancer predisposition syndromes, such as Cowden disease [9,10]. Furthermore, PTEN can modulate immune responses by regulating Fcγ receptor-mediated phagocytosis [11,12].

Human amebiasis is caused by the infection of the enteric protozoan parasite *Entamoeba histolytica*. World Health Organization estimates 50 million people throughout the world suffers from amebic infections, resulting in around 100,000 deaths annually [13]. Infection by *E. histolytica* usually occurs via ingestion of fecally contaminated food or water with the infective cyst of this parasite [14]. Destruction of intestinal epithelial tissue by amoebic trophozoites causes colitis and amoebic dysentery while in some patients trophozoites can infect extraintestinal organs where they form abscesses [15]. It is known that the virulence mechanisms of *E. histolytica* are sustained by actin-associated processes such as migration, adhesion, and trogo-/

phagocytosis as well as vesicular traffic involved in the secretion of proteases [16–19]. A sufficient set of PI-kinases and phosphatases to generate 7 species of phosphoinositides appear to be conserved in *E. histolytica* [20]. AGC kinases have recently been identified as PtdIns(3,4,5) $P_3$-binding proteins and shown to be involved in trogocytosis and phagocytosis in *E. histolytica* [21]. Physiological significance of PtdIns(3)P- and PtdIns(4)P-binding proteins including FYVE domain-containing proteins was shown [22]. In addition, the distinct roles of PtdIns(3) P-binding sorting nexins (SNXs) in trogocytosis have been demonstrated in *E. histolytica* [23]. Among them, PtdIns(3,4,5)$P_3$-mediated signaling is assumed to have a pivotal role in *E. histolytica* virulence. Although physiological roles of PTEN have been well established in higher eukaryotes, the role of PTEN in *E. histolytica* in pathogenesis remains elusive.

In the present study, we characterized the biological roles of EhPTEN1, which shows the highest expression at the transcript level in the trophozoite stage among six putative PTENs encoded by the genome. We have shown that EhPTEN1 is enzymatically active against PtdIns (3,4,5)$P_3$ and is required for optimal growth of *E. histolytica* cells. We have also found that EhPTEN1 is involved in the regulation of different modes of endocytosis, namely fluid-phase endocytosis, receptor-mediated endocytosis, phago-, trogocytosis, and cell migration.

## Results

### Identification and features of *PTEN* genes in *E. histolytica*

A genome-wide survey of PTEN in the genome of *E. histolytica* HM-1:IMSS reference strain (AmoebaDB, http://amoebadb.org) by BLASTP analysis using human PTEN (P60484) as a query, revealed that *E. histolytica* possesses 6 possible PTEN or PTEN-like proteins that contain PTEN phosphatase domain, and show different domain configurations (Fig 1A) [20]. We tentatively designated them firstly in an ascending order of the number of recognizable domains and secondly in a descending order of the overall length (EhPTEN1, EHI_197010; EhPTEN2, EHI_098450; EhPTEN3, EHI_131070; EhPTEN4, EHI_054460; EhPTEN5, EHI_041900; EhPTEN6, EHI_010360). Our previous transcriptome data [24–26] verified that one protein (EhPTEN1, EHI_197010) is highly expressed in the trophozoite stage in both *E. histolytica* HM-1: IMSS cl6 and G3 strains, while the 5 other PTENs are expressed at relatively low levels (Fig 1B). RNA seq data from our laboratory [27] indicate the abundance (FPKM) of six EhPTEN is as follow: EhPTEN1-6, 173.7, 25.8, 9.7, 6.7, 17.8, and 7.0 in HM-1:IMSS cl-6 transfected with HA control vector; 62.0, 11.6, 5.9, 2.0, 9.6, and 7.6 in G3 strain transfected with psAP2-Gunma mock vector. EhPTEN1 shows 39% mutual identity to human PTEN at the amino acid level (S1 Table). Multiple sequence alignment by ClustalW program (http://clustalw.ddbj.nig.ac.jp) shows that the key catalytic residues in the phosphatase domain (H-C-K/R-A-G-K-G-R) needed for lipid and protein phosphatase activity [4,28] are well conserved in EhPTEN1 (Fig 1C). In addition, the PtdIns(4,5)$P_2$-binding motif (K/R-x4-K/R-x-K/R-K/R-R, PDM domain), which is predicted to regulate the recruitment of protein to the plasma membrane, located at the amino terminus, is also conserved in EhPTEN1 [6,29]. The cytosolic localization signal (D-G-F-x-L-D-L, CLS), where mutation of phenylalanine was shown to induce nuclear localization [30], as well as threonine and isoleucine responsible for TI loop formation in an extension of the active site pocket are also conserved (Fig 1C) [28]. EhPTEN1 also possesses C2 domain which has an affinity for membrane phospholipids and helps PTEN to be recruited to the cell membrane [28]. The homology of the first 190 amino acids in the amino terminus of PTEN to the actin binding protein tensin may suggest the involvement of PTEN in the regulation of actin dynamics. InterPro domain search annotates the region of a.a. 550–680 of EhPTEN1 as a domain of unknown function (DUF457). The two PEST sequences, rich in proline, glutamic acid, serine, and threonine, at the C terminus of

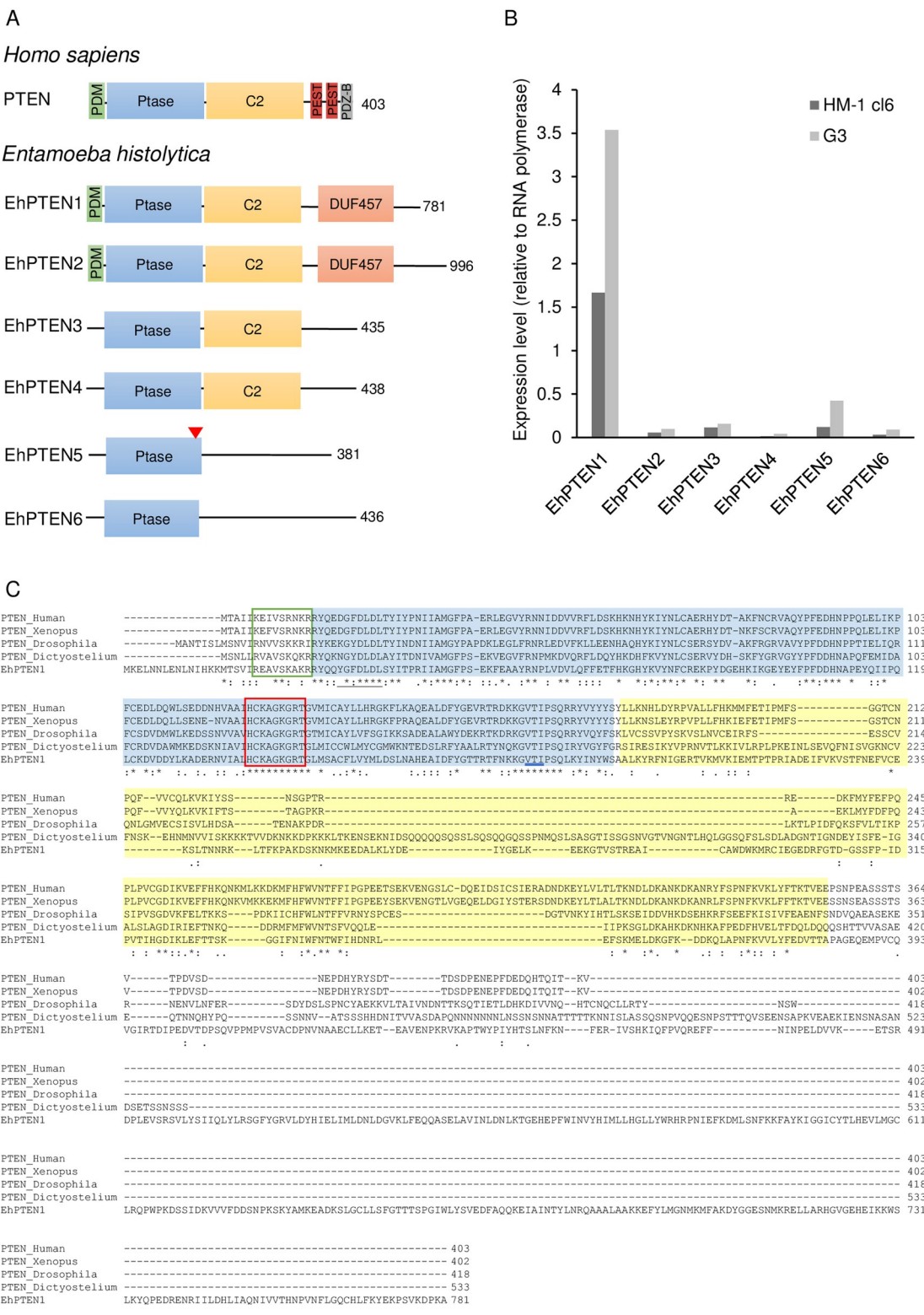

**Fig 1. Structural features and sequence alignments of PTEN in *Entamoeba histolytica*. (A)** Domain organization of PTEN from human and *E. histolytica*. PDM [PtdIns(4,5)P$_2$-binding motif], Ptase (Phosphatase tensin-type domain), C2 (C2 tensin-type domain), DUF547 (Domain of unknown function), PDZ-BM (PDZ-binding motif), PEST (proline, glutamine, serine, threonine sequence), red triangle indicates the nuclear localization sequence. **(B)** Relative mRNA expression of PTEN homologs in *E. histolytica* trophozoites HM1: IMSS cl6 and G3 strains. The steady-state mRNA levels of PTEN isotypes in *E. histolytica* were created

using the data from our previous work [24–26] where transcriptome of trophozoites of *E. histolytica* HM-1:IMSS cl6 reference strain was analyzed using DNA microarrays technology [25]. The levels of steady-state mRNA of PTEN isotypes are shown after normalization against that of RNA polymerase II. **(C)** Multiple amino acid sequence alignment of human PTEN (P60484), *Xenopus laevis* (AAD46165.1), *Drosophila melanogaster* (Q9Y0B6), *Dictyostelium discoideum* PTEN (Q8T9S), and EhPTEN1 (XP_653141.2) was constructed by using clustalw algorithm (http://clustalw.ddbj.nig.ac.jp). PTEN phosphatase domain and C2 domain are shown with blue and yellow backgrounds, respectively. The green rectangle corresponds to the PtdIns(4,5)P$_2$-binding motif (PDM domain). Amino acid residues implicated for PtdIns(3,4,5)P$_3$ catalysis are marked with a red rectangle. Cytosolic localization signal and restudies important for TI loop formation are indicated in black and blue lines, respectively. Note that for human PTEN, only the amino terminal part is shown.

human PTEN, are not conserved in EhPTEN1. Although PEST sequences are known to enhance proteolytic sensitivity, the regulation of EhPTEN1 functions may differ from the human PTEN [31]. EhPTEN1 also lacks PDZ-binding motif (T/S-x-V) located at the C terminus in human PTEN and facilitates the protein-protein interactions [6,32]. The human PTEN was shown to interact with several PDZ domain-containing proteins, such as membrane-associated guanylate kinase inverted (MAGI) and Na$^+$/H$^+$ exchanger regulatory factor (NHERF) in a PDZ-dependent manner [8,9]. Interaction of MAGI to the PTEN C-terminal PDZ domain facilitates targeting of PTEN to the plasma membrane, thereby enhancing its activity [9]. While NHERF regulates the activation of the PI3K–AKT pathway by binding and recruitment of PTEN to platelet-derived growth factor receptor (PDGFR) [8]. However, the *E. histolytica* genome apparently does not encode proteins with similarity to NHERF or MAGI, as indicated by BLAST search. Therefore, these PDZ domain binding proteins may be unique to mammals, which represents a major difference between mammalian PTEN and EhPTEN1. The ~400 a.a. carboxyl-terminal extension which is absent in human ortholog and rich in charged amino acids could be involved in regulating its protein-protein interactions.

## Cellular localization and dynamics of EhPTEN1 in the motile *E. histolytica* trophozoite

To examine the cellular localization of EhPTEN1 in trophozoites, we established a transformant line expressing EhPTEN1 with the GFP-tag at the amino terminus (GFP-EhPTEN1). The expression of GFP-EhPTEN1 or GFP (control) in transformant trophozoites was verified by immunoblot analysis using anti-GFP antibody. A single band corresponding to non-truncated GFP fusion protein with an expected molecular mass of GFP-EhPTEN1 (90 kDa plus 26 kDa for the GFP tag) was observed in the GFP-EhPTEN1-expressing transformant (Fig 2A). Live imaging analysis revealed that GFP-EhPTEN1 was localized throughout the cytosol (Fig 2B and S1 Movie). The line intensity plots across the GFP-EhPTEN1 overexpressing trophozoites further demonstrated the enrichment of GFP-EhPTEN1 along the extended pseudopod of the motile trophozoite (Fig 2B) (Two additional images are also show in S1A and S1B Fig). The normalized average fluorescence intensities at the leading regions of pseudopods were nearly 1.5-fold higher in GFP-EhPTEN1 overexpressing trophozoites compared to GFP-expressing mock transformants (Fig 2C). Furthermore, the intensity line plot of GFP-expressing control strain showed no accumulation of GFP signal on pseudopods (S2 Fig and S2 Movie). These findings confirm the enrichment of GFP-EhPTEN1 in the pseudopod-like protrusive structures. Although the levels of expression of GFP-EhPTEN1 and GFP varied in the respective population, the accumulation of GFP-EhPTEN1 in the pseudopod like structure at the initial stage of trogocytosis and phagocytosis (S4 Movie) was observed even in the cell that expressed GFP-EhPTEN1 at low levels (showing only dim fluorescence). Similarly, immunofluorescence imaging of HA-EhPTEN1 overexpressing trophozoites using anti-HA antibody revealed that HA-EhPTEN1 was localized in the cytoplasm in steady-state, and enriched in

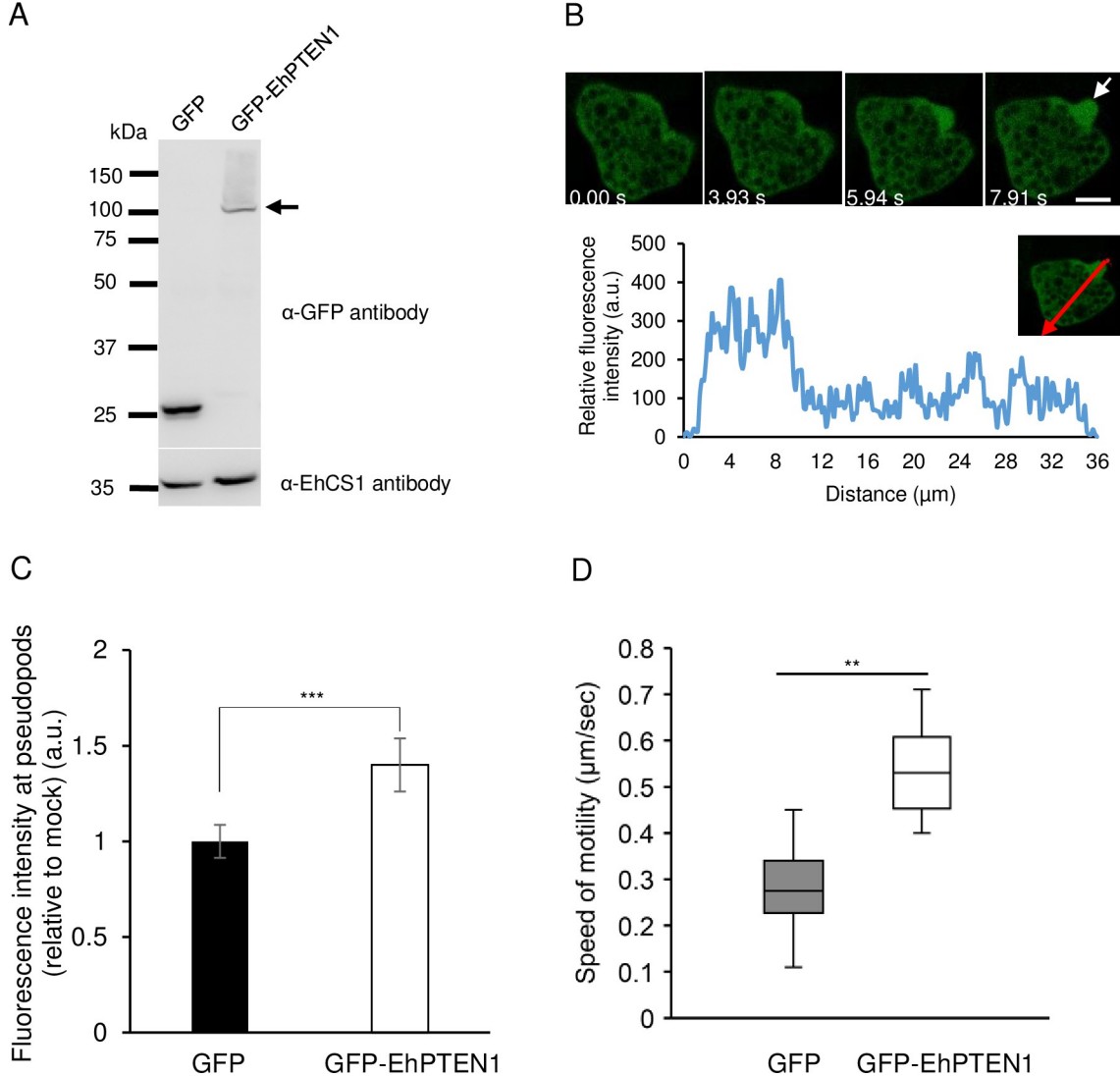

**Fig 2. Expression and localization of GFP-EhPTEN1 in motile trophozoites. (A)** Immunoblot of GFP-EhPTEN1 and GFP (control) in *E. histolytica* transformants. Approximately 30 μg of total lysates from mock-transfected control (GFP) and GFP-EhPTEN1-expressing transformant (GFP-EhPTEN1) were subjected to SDS-PAGE and immunoblot analysis using anti-GFP antibody and anti-CS1 antibody. Arrow indicates GFP-EhPTEN1. **(B)** Live imaging montage showing a time series of motile trophozoites expressing GFP-EhPTEN1. The pseudopodal localization of GFP-EhPTEN1 is indicated by white arrow. The line intensity plot shows GFP-EhPTEN1 intensity in pseudopods vs. cytoplasm with the distance. (Scale bar, 10 μm) **(C)** Relative fluorescence intensities were quantified in the pseudopod regions of GFP-EhPTEN1 and GFP control expressing trophozoites then normalized to the fluorescence intensities in the total cells. Data points in the graph show the mean and error bars represent standard deviation for 30 cells. Statistical significance was examined with t-test (***P<0.001). **(D)** Cell motility of GFP and GFP-EhPTEN1 transfected strains. Time-lapse images of the transformant trophozoites were collected every sec for 5 min using CQ1 and 30 cells were selected randomly for analysis by CellPathfinder software. The average of three independent experiments is shown. Statistical significance was examined with Dunnet test (**P< 0.05).

pseudopods when the fluorescence signal across the line traversing the cell was measured (S3C Fig). HA-EhPTEN1 localization was also evaluated by measuring the fluorescence signal in the different portion of the cell, confirming that HA-EhPTEN1 is highly concentrated in the pseudopods, compared to other parts of the cell (S3D Fig). The migration (motility) of the GFP-EhPTEN1 overexpressing trophozoites using the montage of time-lapse imaging was 0.54±0.09 μm/sec (mean±S.D.), which was significantly greater than that of control GFP

expressing transformants (0.27±0.08 μm/sec) (Fig 2D). We also investigated the effect of repression of *EhPTEN1* gene expression and found that *EhPTEN1* gene silencing reduced migration (see below).

## Localization of EhPTEN1 during trogocytosis and phagocytosis

The fact that EhPTEN1 was previously identified as a PtdIns(3)P-binding effector and suggested to be involved in the phagosome biogenesis [23], prompted further characterization of the role of EhPTEN1 in host cell internalization. To examine the role of EhPTEN1 in ingestion of mammalian cells, we first examined trogocytosis (i.e., nibbling or chewing of a part of a live cell) of Chinese hamster ovary (CHO) cells by GFP-EhPTEN1 and GFP expressing transformant lines. We co-cultured trophozoites of the two transformant lines with live CHO cells that had been stained with CellTracker Orange. Time-lapse imaging of trogocytosis of CHO cells by the amoebae (Fig 3A and S3 Movie) revealed that GFP-EhPTEN1 was accumulated in the region that covers, but not always in close proximity to, the tunnel-like structure, which is the extended neck (or tube)-like structure connecting the enclosed (or being enclosed) trogosome and the remaining portion of the target cell that is partially ingested (Fig 3A–3D). Upon completion of closure of the trogosome, GFP-EhPTEN1 appeared to be dissociated from the region around the trogosome and the tunnel-like structure (Fig 3A–3E). The quantification of the fluorescence intensity in a cross section of the cell confirmed the dynamism of GFP-EhPETN1 during trogocytosis (Fig 3B–3E). In contrast, at the very early phase of trogocytosis, GFP-EhP-TEN1 was not concentrated on the newly formed trogocytic cup.

The dynamics of GFP-EhPTEN1 in a course of phagocytosis (i.e., internalization with a single bite, not multiple bites) of dead CHO cells was also examined. The live imaging of GFP-EhPTEN1 expressing trophozoites co-cultured with pre-killed CHO cells (Fig 4 and S4 Movie) showed an enrichment of GFP-EhPTEN1 at the tip of the leading edge of the phagocytic cup during the internalization of dead host cells until phagosome closure (Fig 4A and 4B). Soon after closure of the phagosome, GFP-EhPTEN1 was concentrated on the closing side of the phagosome (Fig 4A–4C), and rapidly disappeared soon after (Fig 4D). The fluorescence intensity line plot of a cross section (as indicated by arrows) of the cell also reinforced the observation (Fig 4A–4D). As control, GFP-expressing mock strain showed no observable concentration of GFP signal in a course of CHO ingestion (S4 Fig and S5 Movie).

## Effect of overexpression of EhPTEN1 on trogocytosis and phagocytosis

The dynamism of GFP-EhPTEN1, as revealed by live imaging, suggests that EhPTEN1 plays a role in the early to middle stages of trogo- and phagocytosis. We examined the effect of GFP-EhPTEN1 overexpression on the efficiency (i.e., speed and volume of internalization of prey) of trogocytosis and phagocytosis. GFP-EhPTEN1 expressing and GFP-expressing mock transformant strains were incubated with either live or pre-killed CHO cells that had been stained with CellTracker Orange to allow trogocytosis or phagocytosis, respectively. Internalization of CHO cells by the amoebae was measured by CQ1 confocal quantitative image cytometer (Figs 5 and 6). Three parameters were measured and compared between GFP-EhPTEN1 expressing and GFP-expressing mock transformant strains: the number of CHO-containing trogosomes or phagosomes per ameba (Figs 5A and 6A), the volume of all CHO-containing trogosomes or phagosomes per ameba (Figs 5B and 6B), and the percentage of the amebae that ingested CHOs (Figs 5C and 6C). GFP-EhPTEN1 overexpression caused statistically significant reduction in all three parameters in both trogocytosis and phagocytosis.

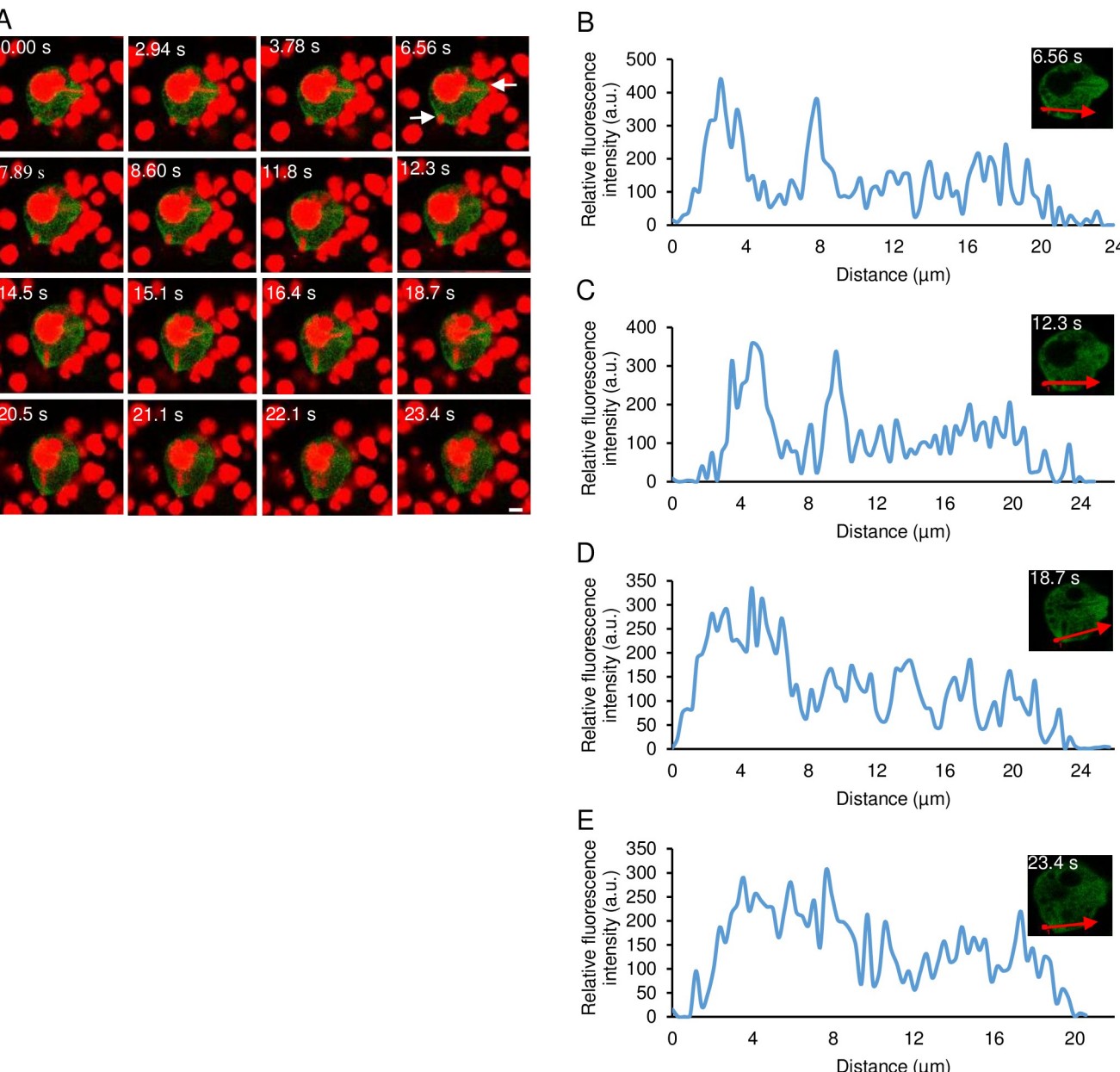

**Fig 3. Localization of GFP-EhPTEN1 during trogocytosis. (A)** Time series montage showing the localization of GFP-EhPTEN1 during trogocytosis of live CHO cells by amoebic trophozoites. The site of trogocytosis is marked with arrow. (Scale bar, 10 μm). **(B)** Analysis of GFP-EhPTEN1 intensity along the line drawn at the initial phase of CHO internalization soon after attachment. **(C)** The plot showing the intensity of GFP-EhPTEN1 along the line drawn reveals its enrichment in the tunnel formed during amoebic trogocytosis. **(D)** The graph shows the intensity of GFP-EhPTEN1 at the late phase of trogocytosis soon after closure of the trogocytic cup **(E)** The graph shows the intensity of GFP-EhPTEN1 after the closure of the trogocytic cup.

## Gene silencing of *EhPTEN1* enhances trogocytosis and phagocytosis in *E. histolytica*

Conversely, we attempted to verify if repression of *EhPTEN1* gene expression by antisense small RNA-mediated transcriptional gene silencing [33] causes reverse phenotypes: enhancement of trogocytosis and phagocytosis. Small antisense RNA-based transcriptional gene silencing, used in this study, was robustly usable only in G3, but not in its parental strain HM-1:

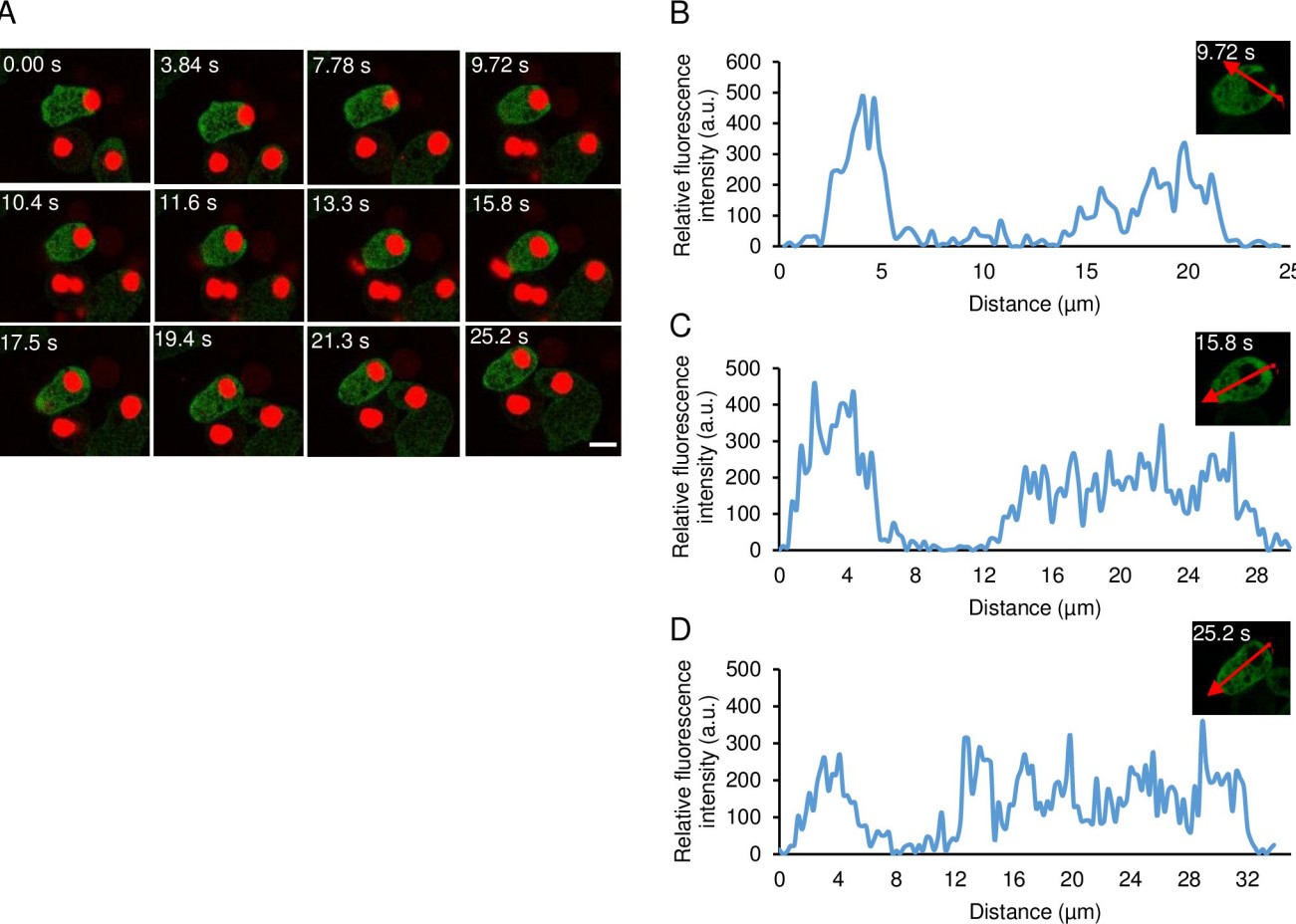

**Fig 4. Localization of GFP-EhPTEN1 phagocytosis of pre-killed CHO cells. (A)** Montage of live trophozoite expressing GFP-EhPTEN1 ingesting pre-killed CHO cells by phagocytosis. (Scale bar, 10 μm). **(B)** Analysis of intensity of GFP-EhPTEN1 across the phagocytic cup along the line drawn. **(C)** The plot showing the intensity of GFP-EhPTEN1 along the line drawn across the newly formed phagosome. **(D)** The graph shows the intensity of GFP-EhPTEN1 after phagosome maturation.

IMSS [33]. The silencing of the *EhPTEN1* gene expression was confirmed by RT-PCR and the level of silencing was estimated to be approximately 77.0±9.2% compared to the mock control (G3 transfected with the empty psAP2-Gunma vector) (Fig 7A and 7B). Non-specific off-target gene silencing of other *EhPTEN* genes (*EhPTEN2-6*) was ruled out, except for *EhPTEN2*, which showed a slight reduction in *EhPTN1* gene silenced strain, validating gene-specific silencing (S5 Fig). The *RNA pol II* transcript level was also unaffected. *EhPTEN1* gene silenced and psAP2 mock transformants were cultivated with live or dead CHO cells and images were captured every 10 min for 1 hr. As expected, *EhPTEN1* gene silenced strain showed an enhancement of trogocytosis and phagocytosis (Fig 8). All three parameters to evaluate trogo-cytosis and phagocytosis, as above, i.e., the number of CHO-containing trogosomes or phago-somes per ameba (Fig 8A and 8B), the volume of all CHO-containing trogosomes or phagosomes per ameba (Fig 8C and 8D), and the percentage of the amebae that ingested CHOs (Fig 8E and 8F) was significantly increased in *EhPTEN1*gene silenced strain compared to the psAP2 mock control stain. For instance, the volume of trogosomes and phagosomes increased by around 1.4-fold for trogocytosis and 2-fold for phagocytosis, respectively, in *EhP-TEN1* gene silenced strain at later time points of coincubation (at 40–60 mins). Together with

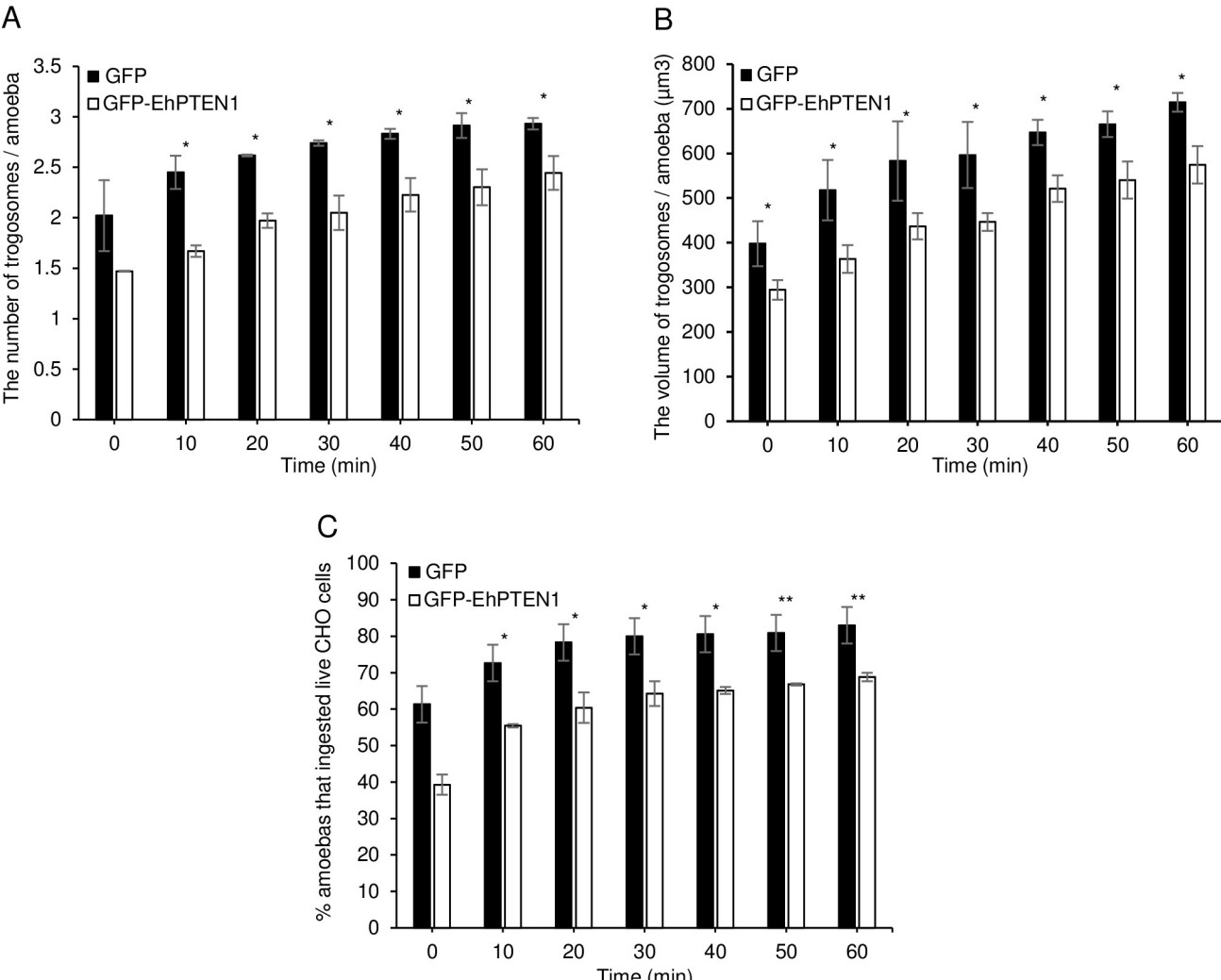

**Fig 5. Effect of GFP-EhPTEN1 expression on trogocytosis. (A)** Trophozoites of GFP mock transfected and GFP-EhPTEN1 expressing strains were incubated with live CHO cells that have been stained with CellTracker Orange to evaluate trogocytosis. The images were taken on CQ1 as described in Materials and methods and analyzed to calculate the average numbers of CHO cell-containing trogosomes per amoeba. **(B)** The volume of the ingested CHO cells was calculated using three-dimensionally reconstituted data. **(C)** The percentage of amoeba trophozoites that ingested live CHO cells. Experiments were conducted three times independently in triplicates and a representative data set is shown. Statistical significance was examined with t-test ($^*$P<0.05, $^{**}$P<0.01). Error bars indicate standard deviations of two biological replicates.

the results of EhPTEN1 overexpression, shown above, these data indicate that EhPTEN1 serves as a negative regulator of trogocytosis and phagocytosis.

## EhPTEN1 is a positive regulator for the fluid-phase and receptor-mediated endocytosis in *E. histolytica*

To investigate the role of EhPTEN1 in pinocytosis of the fluid-phase marker and receptor-mediated endocytosis, we examined the internalization of RITC dextran and transferrin. Pinocytosis was analyzed by measuring, on a fluorometer, the fluorescence intensity of fluid-phase marker, RITC dextran, which was internalized after incubation of amoebic transformants with RITC dextran at 35˚C for up to 1 hr. Overexpression of GFP-EhPTEN1 cause approximately 30% increase in pinocytosis in comparison to GFP mock control (53±5.3 or 28±13% at time 30

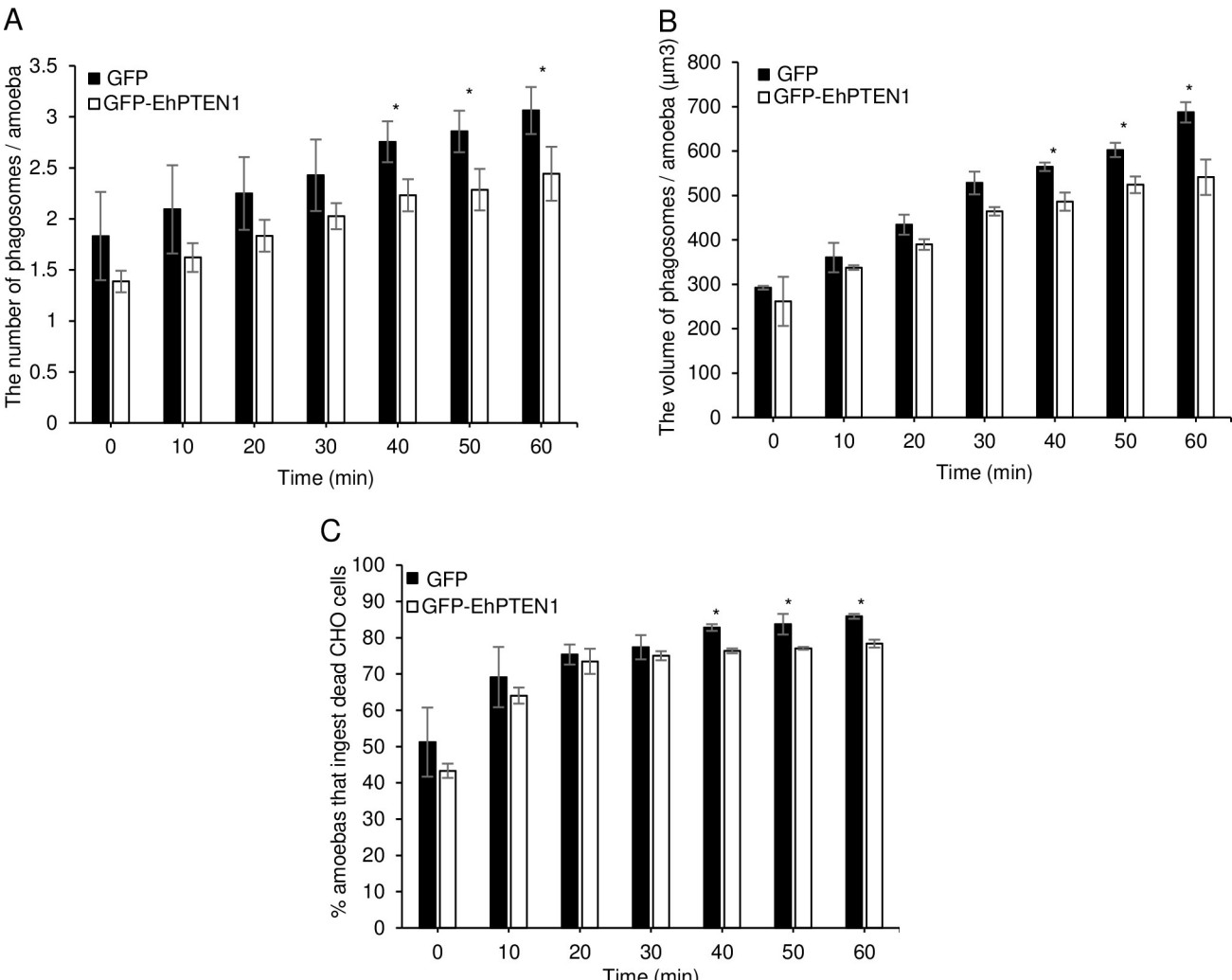

**Fig 6. Effect of GFP-EhPTEN1 expression on phagocytosis. (A)** Trophozoites of GFP mock transfected and GFP-EhPTEN1 expressing strains were incubated with heat killed CHO cells that have been stained with CellTracker Orange to evaluate phagocytosis. The images were taken on CQ1 as described in Materials and methods and analyzed to calculate the average numbers of CHO cell-containing phagosomes per amoeba. **(B)** The volume of the ingested CHO cells was calculated using three-dimensionally reconstituted data. **(C)** The percentage of amoeba trophozoites that ingested pre-killed CHO cells. Experiments were conducted three times independently in triplicates and a representative data set is shown. Statistical significance was examined with t-test ($^*$P< 0.05). Error bars indicate standard deviations of two biological replicates.

or 60 min, respectively; p<0.05, Fig 9A). Conversely, *EhPTEN1* gene silenced strain showed an approximately 30% decrease in pinocytosis at 30–60 min, as compared to psAP2 mock control cells (30±8.2 or 25±5.1% decrease at time 30 or 60 min, respectively, p<0.05, Fig 9B).

We next examined internalization of transferrin conjugated with AlexaFluor 568 by CQ1 image cytometer. Transferrin is presumed to be internalized via receptor-mediated endocytosis. The volume of endosomes that contained transferin-AlexaFluor 568 increased by 30–50% in GFP-EhPTEN1 overexpressing trophozoites compared to GFP mock control at all time points up to 1 hr (p<0.05, Fig 10A). Conversely, transferrin endocytosis decreased by 30–40% in *EhPTEN1* gene silenced strain compared to the psAP2 mock strain at 40–60 mins (p<0.05, Fig 10B). These data indicate that EhPTEN1 positively regulates pinocytosis of the fluid-phase maker and receptor-mediated endocytosis in *E. histolytica*.

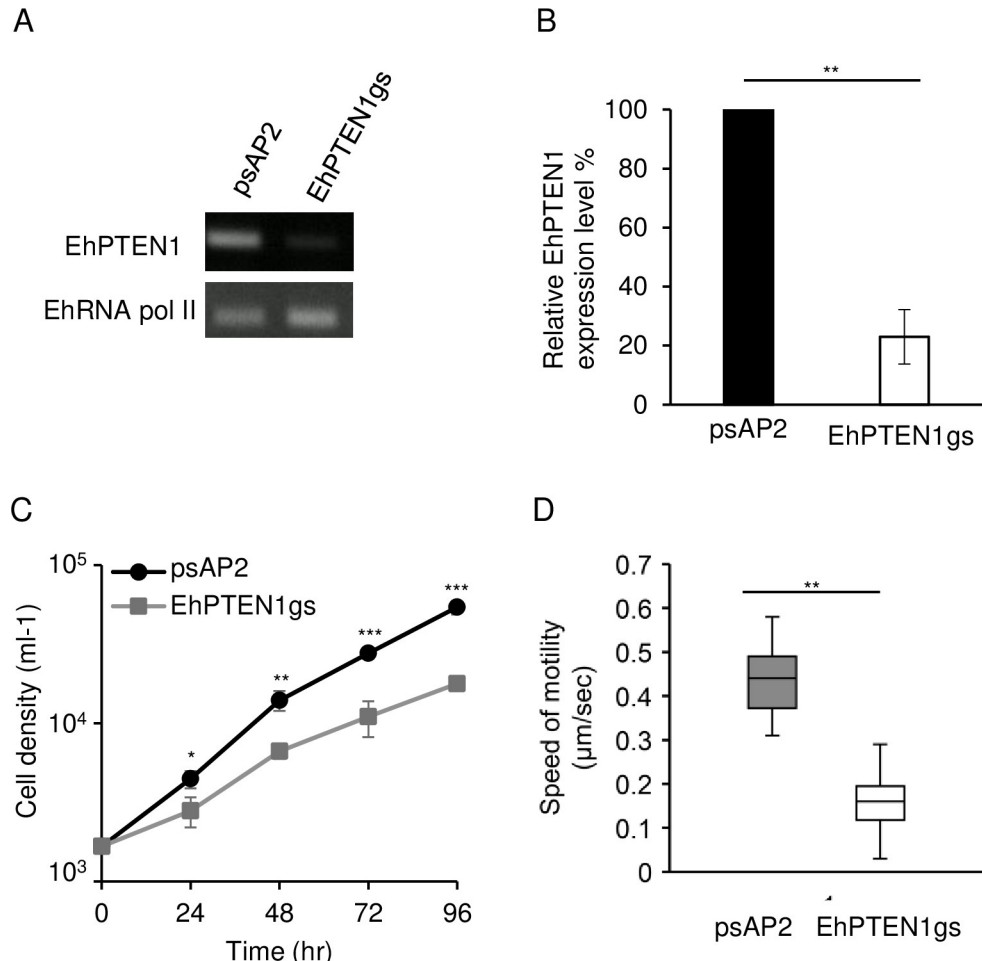

**Fig 7. Establishment and phenotypes of EhPTEN1 gene silenced strain. (A)** Confirmation of gene silencing by RT-PCR analysis of psAP2-Gunma mock transfected and *EhPTEN1* gene silenced strain (EhPTEN1gs) strain. Transcripts of *EhPTEN1* and RNA polymerase II genes were amplified by RT-PCR from cDNA isolated from the transformants and examined by agarose gel electrophoresis. **(B)** Relative levels of EhPTEN1 transcripts by qRT-PCR analysis in EhPTEN1gs and psAP2 mock strains. The transcript levels were normalized against RNA polymerase II and are shown in percentage relative to the transcript level in mock control strain. Data shown are the means ± standard deviations of two biological replicates. Statistical comparison is made by t-test (**P<0.01). **(C)** Growth kinetics of psAP2 mock and EhPTEN1gs transformants s during 96 h incubation in BI-S-33 medium. Data shown are the means ± standard deviations of three biological replicates. Statistical comparison is made by t-test (*P<0.05, **P<0.01, ***P<0.001). **(D)** Cell motility of psAP2 mock transfected and EhPTEN1 gene silenced strains. The indicated transformant trophozoites were pre-stained with CellTracker green and time-lapse images were collected every sec for 2 min using CQ1 and 30 cells were selected randomly for analysis by CellPathfinder software. The experiments were performed three times independently. The average of three independent experiments is shown. Statistical significance was examined with Dunnet test (**P<0.05).

## EhPTEN1 is essential for optimum growth and migration of *E. histolytica*

The biological role of EhPTEN1 in trogo-, phagocytosis, and endocytosis was clearly demonstrated as above. To investigate other physiological roles of EhPTEN1 in *E. histolytica*, the growth kinetic was monitored in *EhPTEN1* gene silenced and control strains. *EhPTEN1* gene silencing caused significant growth defect: the population doubling time of *EhPTEN1* gene silenced and control strains was 28.1±0.41 and 19.1±0.52 hr, respectively (P<0.05; Fig 7C). We next examined the migration of the trophozoites of *EhPTEN1* gene silenced and control strains

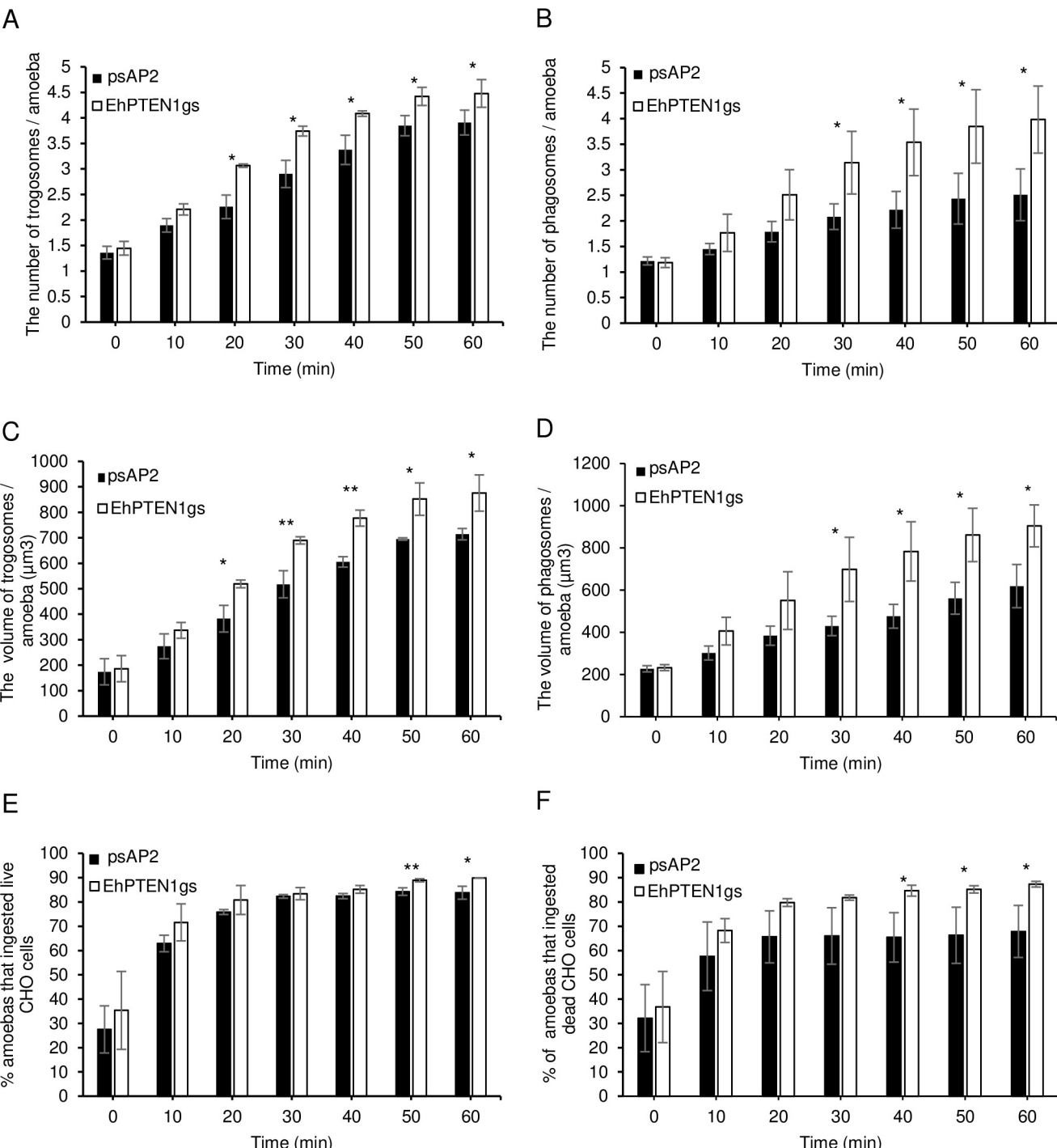

**Fig 8. The effect of gene silencing of EhPTEN1 on trogocytosis and phagocytosis. (A, C, and E)** Trophozoites of psAP2 mock and EhPTEN1gs strains were prestained with CellTracker Blue were incubated with live CHO cells that have been stained with CellTracker Orange to evaluate trogocytosis. The images were taken on CQ1 as described in Materials and methods. **(B, D, and F)** Trophozoites of psAP2 mock and EhPTEN1gs strains were prestained with CellTracker Blue were incubated with heat killed CHO cells that have been stained with CellTracker Orange to evaluate phagocytosis. The images were taken on CQ1 as described in Materials and methods. **(A)** The average numbers of CHO cell-containing trogosomes per amoeba. **(B)** The average numbers of CHO cell-containing phagosomes per amoeba. **(C)** The volume of the trogosomes were calculated using three-dimensionally reconstituted data. **(D)** The volume of the phagosomes were calculated using three-dimensionally reconstituted data. **(E)** The percentage of amoeba trophozoites that ingested live CHO cells. **(F)** The percentage of amoeba trophozoites that ingested pre-killed CHO cells. Experiments were conducted three times independently in triplicates. Statistical significance was examined with t-test ($^*$P<0.05, $^{**}$P<0.01). Error bars indicate standard deviations of three biological replicates.

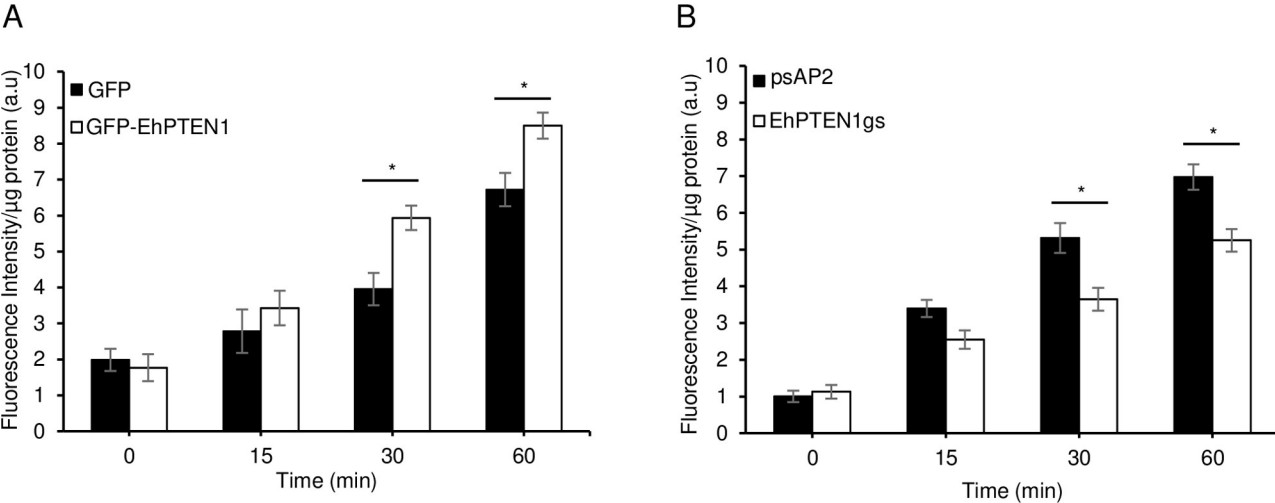

**Fig 9. Effect of EhPTEN1 on pinocytosis. (A)** The effect of GFP-EhPTEN1 expression on pinocytosis. Trophozoites of GFP mock transfected and GFP-EhPTEN1 expressing strains were assayed for RITC dextran uptake in a time-dependent manner. **(B)** The effect of pinocytosis upon EhPTEN1 silencing in comparison to psAP2 mock control. Trophozoites of mock and EhPTEN1gs strains were incubated in BI-S-33 medium containing RITC dextran and assayed for its uptake for indicated time points. Experiments were conducted three times independently and statistical significance was examined with t-test ($^*$P<0.05). Error bars indicate standard deviations of three biological replicates.

using time lapse imaging by CQ1. The velocity of motility was >60% reduced in *EhPTEN1* gene silenced strain (0.16±0.07 μm /min) compared to the psAP2 mock control (0.44±0.08 μm /min) (Fig 7D).

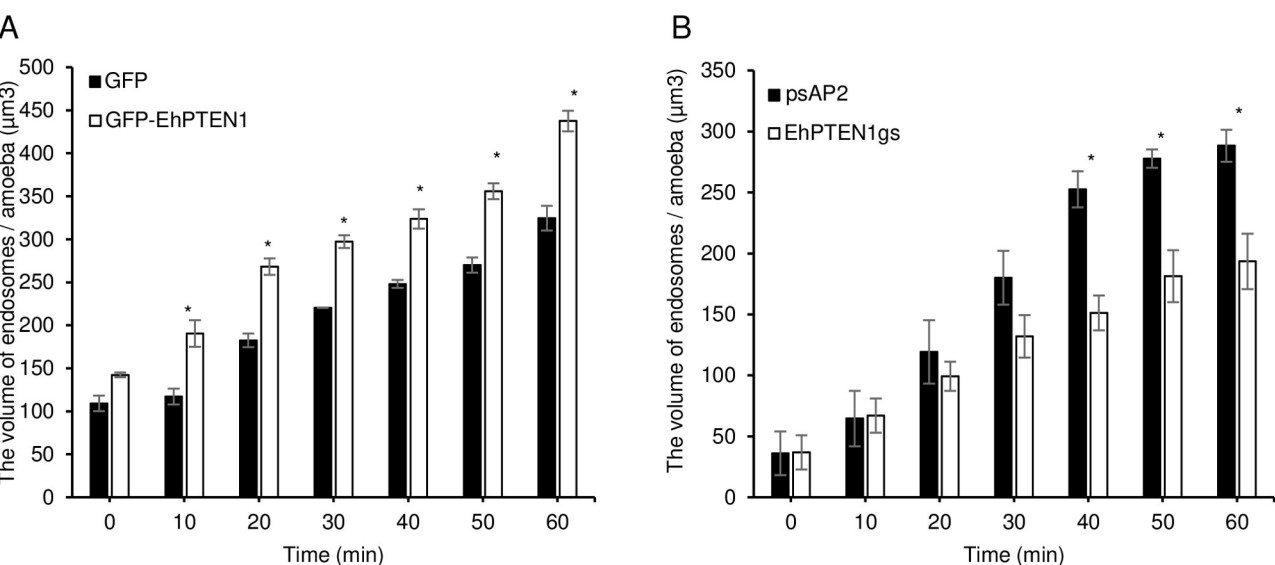

**Fig 10. Effect of EhPTEN1 on endocytosis. (A)** The effect of GFP-EhPTEN1 expression on endocytosis. Trophozoites of GFP mock transfected and GFP-EhPTEN1 expressing strains were incubated in BI-S-33 medium containing transferrin and images were taken every 10 min for 1 hr by CQ1 as described in Materials and methods. The volume of endosomes was calculated using three-dimensionally reconstituted data. **(B)** The effect of EhPTEN1 gene silencing on endocytosis. Images of psAP2 mock and EhPTEN1gs transformant trophozoites that have been co-cultivated with transferrin were taken every 10 min for 1 hr by CQ1 as described in Materials and methods. The volume of endosomes was calculated using three-dimensionally reconstituted data. All experiments were conducted three times independently and statistical significance was examined with t-test ($^*$P<0.05). Error bars indicate standard deviations of three biological replicates.

## Demonstration of phosphatase activity and substrate specificity of EhPTEN1

To see if EhPTEN1 possesses lipid phosphatase activity, bacterial recombinant EhPTEN1 with the histidine tag at the amino terminus was produced using the pCold I *E. coli* expression system. SDS-PAGE analysis followed by Coomassie Brilliant Blue staining showed that the purified recombinant EhPTEN1 was apparently homogenous with the predicted molecular mass of 96 kDa including the histidine tag (S6A Fig). Immunoblot analysis of the purified recombinant protein using His-tag antibody confirmed the absence of truncation (S6B Fig). We first examined the enzymatic activities of recombinant EhPTEN1 using a variety of phosphoinositides (PIs) as substrates. EhPTEN1 revealed reasonable activity in a broad pH range with maximum activity obtained at pH 6.0 when the reaction was performed with 50 µM PtdIns(3,4,5)$P_3$ at 37˚C for 40 min (S6C Fig). We then determined the substrate specificity of EhPTEN1, using a panel of di-C8 PIs. EhPTEN1 showed highest activity with PtdIns(3,4,5)$P_3$ with the apparent specific activity of 8.18±0.78 nmol/min/mg (Fig 11). EhPTEN1 also catalyzed dephosphorylation of PtdIns(3,4)$P_2$ and PtdIns(3,5)$P_2$ with 6- or 3- fold lower specific activities, respectively, compared to that toward PtdIns(3,4,5)$P_3$. The activities against PI monophosphates and PtdIns(4,5)$P_2$ were relatively low. All these characteristics are similar to those of human PTEN [34,35]. A comparison of kinetic parameters of EhPTEN1 reveals a higher affinity towards PtdIns(3,4,5)$P_3$ ($K_m$ = 92.5 ± 4.72 µM) as compared to PtdIns(3,4)$P_2$ (292 ± 18.8 µM) and PtdIns(3,5)$P_2$ (161 ± 20.1 µM) demonstrating that in vitro is as well the preferred substrate (Table 1 and S8 Fig). Furthermore, we produced a mutant form of EhPTEN1, in which the cysteine residue implicated for catalysis is replaced with serine by site-directed mutagenesis (EhPTEN1_C140S) (S10 Fig). Recombinant EhPTEN1_C140S showed markedly lower phosphatase activity towards PtdIns(3,4,5)$P_3$ when compared to wild type, with the low specific activity (0.77 ± 0.14 nmol/min/mg). Moreover, expression of the inactive mutant GFP-EhPTEN1_C140S did not cause significant changes in pinocytosis, trogocytosis, and

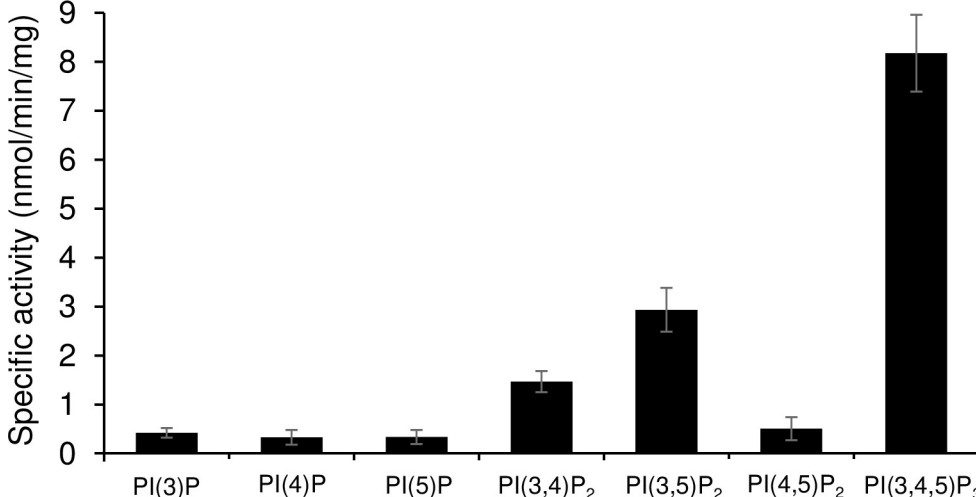

**Fig 11. Substrate specificity and enzymatic activity of EhPTEN1.** Determination of EhPTEN1 Specific Activity. The specific activity of bacterial recombinant EhPTEN1 fusion protein toward a panel of synthetic di-C8-phosphoinositide substrates was determined using a malachite green-based assay for inorganic phosphate. Reactions were carried out in a volume of 25 µl for 40 min at 37$^\circ$C, then terminated by the addition of 100 µl of malachite green reagent as described in Materials and methods. The absorbance at 630 nm was measured and phosphate released was quantified by comparison to a standard curve of inorganic phosphate. The means ± standard deviations of three independent experiments performed in duplicates are shown.

**Table 1. Kinetic parameters of EhPTEN1.**

| Substrate | Km (μM) | Vmax (nmoles min$^{-1}$ mg$^{-1}$) | Kcat (min$^{-1}$) |
|---|---|---|---|
| PI(3,4)P$_2$ | 292 ± 18.8 | 6.02 ± 1.11 | 0.11 ± 0.02 |
| PI(3,5)P$_2$ | 161 ± 20.1 | 8.40 ± 0.42 | 0.15 ± 0.01 |
| PI(3,4,5)P$_3$ | 92.5 ± 4.72 | 16.9 ± 1.83 | 0.31 ± 0.03 |

Assay was performed as described in Materials and methods in the presence of MOPS, EhPTEN1, and PtdInsPs. Reaction was conducted at 37˚C at pH 6.0. Mean ± SEM of duplicates are shown.

phagocytosis of *E. histolytica* trophozoites, which indicates that the enzymatic activity of EhPTEN1 is responsible for its phenotypes (S9 Fig).

## Demonstration of phospholipid binding of EhPTEN1

The lipid overlay assay using amebic lysates from GFP-EhPTEN1 expressing and GFP-expressing mock transformants showed that EhPTEN1 preferentially bound to PtdIns(3)P, PtdIns(4)P, PtdIns(5)P, and, to a lesser extent, PtdIns(3,5)P$_2$, and PtdIns(4,5)P$_2$ (S7 Fig). Furthermore, recombinant His-tagged EhPTEN1 also revealed a similar binding affinity toward a panel of PIs on the membrane, which is similar to the data given for recombinant human PTEN [36]. The binding of EhPTEN1 with diverse immobilized phospholipids may regulate its localization to different cellular compartment where EhPTEN1 exerts its phosphatase activity as has been shown previously that the binding of PtdIns(3)P with C2 domain of human PTEN target its localization to endosomal membranes [36].

## Discussion

PTEN regulates fundamental roles in higher eukaryotes including cell survival, metabolic changes, cell polarity, and migration [9]. In this study, we have characterized the pivotal functions of EhPTEN1 in migration, endocytosis, and cellular proliferation in *E. histolytica*. Confocal live imaging demonstrated the involvement of EhPTEN1 in the initial and intermediate stages of trogocytosis and phagocytosis. In trogocytosis of a live mammalian cell, EhPTEN1 was enriched in the region where the trogocytic tunnel was newly formed. Similarly, EhPTEN1 was accumulated on the cell periphery close to the leading edge of the phagocytic cup during the internalization of a dead host cell. The recruitment of EhPTEN1 was transient as it gradually became dissociated from the region after the completion of ingestion. These results are in line with the previous study that showed PTEN was associated with forming IgG conjugated zymosan containing phagosomes but disappeared once particle ingestion was completed [37]. The biochemical analysis showed that GFP-EhPTEN1 overexpression caused a reduction in trogocytosis and phagocytosis. In good agreement with these results, knockdown of EhPTEN1 caused remarkable enhancement in phagocytosis of dead CHO cells while trogocytosis toward live CHO cells was slightly increased. These results match those observed in earlier studies where PTEN deficient macrophages displayed enhanced phagocytic ability both in vitro and in vivo, while overexpression of PTEN significantly inhibited phagocytosis in macrophages [12,38–40]. It was previously demonstrated that the depletion of PTEN in macrophages resulted in elevated PtdIns(3,4,5)P$_3$ levels, leading to activation of Vav1 and subsequent activation of Rac1 GTPase, the latter of which induces F-actin polymerization, which in turn enhances the engulfment of targeted cells [41]. While PIPs-mediated signaling and downstream effector in *E. histolytica* is not yet well understood, it has been recently reported that two AGC kinases from *E. histolytica* have the ability to bind PtdIns(3,4,5)P$_3$ and are involved in a panel of endocytic events including trogo-, phago-, and pinocytosis [21]. In

addition, it has also been shown that PtdIns(4,5)P$_2$ is localized on the plasma membrane whereas PtdIns(3,4,5)P$_3$ is localized on the phagocytic cup and the extended pseudopodia in *E. histolytica* trophozoites [42,43]. Furthermore, it was previously reported that treatment with wortmannin, the PI3K inhibitor, impaired phagocytosis of several particles including bacteria, mucin-coated beads, and hRBCs by *E. histolytica* trophozoites [42,44]. These data may be consistent with the mutual antagonistic role of PTEN and PI3K. These observations suggest that the control of PtdIns(3,4,5)P$_3$ synthesis and decomposition are important for the regulation of endocytic events in *E. histolytica*. We have clearly demonstrated phosphatase activity and preferred substrate specificity toward PtdIns(3,4,5)P$_3$ of EhPTEN1. Hence, it is highly conceivable that EhPTEN1 can regulate the local concentrations of PtdIns(4,5)P$_2$ and PtdIns(3,4,5)P$_3$ at the target sites during trogo- and phagocytic processes. It seems conceivable that EhPTEN1 negatively regulates trogo- and phagocytosis by reducing the local PtdIns(3,4,5)P$_3$ concentration, leading to the suppression of actin-dependent cytoskeletal reorganization needed for trogo- and phagocytosis. Indeed, the concentration of EhPTEN1 is swiftly reduced on and close to trogo- and phagocytic cups (not-yet-enclosed) and trogosomes and phagosomes (enclosed) soon after the completion of ingestion.

We have shown that EhPTEN1 is involved in receptor-mediated endocytosis and macropinocytosis of the fluid-phase marker in an opposite fashion as in trogo- and phagocytosis. GFP-EhPTEN1 overexpression enhanced transferrin uptake while *EhPTEN1* gene silencing decreased it. It was shown that at least two concentration-dependent mechanisms for transferrin endocytosis exist in *E. histolytica* [45]: Receptor-mediated endocytosis active at low transferrin concentrations [46] and receptor-independent internalization at high transferrin concentrations [47]. As previously demonstrated, receptor-mediated endocytosis of transferrin in *E. histolytica* is indeed clathrin-mediated [clathrin-mediated endocytosis (CME)] [48], and receptor-mediated endocytosis is in general clathrin-mediated and actin independent [49]. Unlike trogo- and phagocytosis, CME is also distinct in that it depends on PtdIns(4,5)P$_2$ and does not require PtdIns(3,4,5)P$_3$ [49,50]. PtdIns(4,5)P$_2$ binds and recruits several proteins associated with CME formation thus depleting cells of PtdIns(4,5)P$_2$ prevents transferrin receptor endocytosis [1,49–51]. Taken together, EhPTEN1 possibly facilitates the transferrin internalization through augmentation of PtdIns(4,5)P$_2$ synthesis. Furthermore, EhPTEN1 showed similar phenotypes toward pinocytosis of the fluid-phase marker. As stated above, transferrin, when present at high concentrations, is internalized by receptor independent fashion in *E. histolytica* [47]. Thus, it is consistent with our observation that *E. histolytica* internalization of the fluid-phase marker and transferrin by actin-dependent macropinocytosis (S7 and S8 Movies), also as previously shown [52]. We also previously showed that EhAGCK2, which preferentially binds PtdIns(3,4,5)P$_3$ over PtdIns(4,5)P$_2$, is involved in pinocytosis of the fluid-phase marker [21], supporting the actin-dependence of macropinocytosis. However, it was shown that the local production of PtdIns(4,5)P$_2$ in the early stages of macropinocytosis is essential for the formation of ruffles and is partly responsible for the remodeling of the actin cytoskeleton [53]. Furthermore, it has been reported previously that deletion of PTEN in *D. discoideum* caused a reduction in fluid uptake [54]. Nevertheless, the lipid rafts in the plasma membrane of *E. histolytica* is highly enriched with PtdIns(4,5)P$_2$ [43] and disruption of lipid rafts with cholesterol-binding agents significantly inhibited fluid-phase pinocytosis of *E. histolytica* [55]. Altogether, we assume that EhPTEN1 accelerates transient synthesis of PtdIns(4,5)P$_2$ on the plasma membrane which in turn facilitates the formation of actin-associated macropinocytic cup. This study has provided the first observation that PTEN is differentially involved in multiple actin-related cytoskeletal activities.

We have shown that repression of gene expression of *EhPTEN1* caused significant growth defect. This phenotype can be possibly explained by reduced ability in nutrient uptake where macropinocytosis is the primary and widely used method for feeding in amoebae trophozoites

[56]. Furthermore, it was previously demonstrated that the growth defect in *E. histolytica* in low iron medium was rescued by the addition of iron-loaded holo-transferrin, and that holo-transferrin was recognized by an amoebic transferrin receptor and endocytosed via clathrin-coated vesicles [46,57]. These data, taken together, underscore the importance of macropino-cytosis and transferrin endocytosis for the proliferation of amoebae. On the other hand, it was previously shown that loss of PTEN significantly lowered growth in *D. discoideum*, possibly attributable to mislocalization of myosin II during cytokinesis [58,59]. Similarly, myosin II mutants caused reduction in growth and multinucleation in *E. histolytica* [60]. These observations likely support the premise that EhPTEN1 regulates amoebic cell proliferation by regulation of cytokinesis and/or nutrient uptake by macropinocytosis. Furthermore, the requirement of EhPTEN1 for optimum proliferation indicates that *E. histolytica* apparently does not possess compensatory mechanisms for the PIPs dysregulation caused by the loss of EhPTEN1, and thus have posed it as rational drug target.

We have shown that GFP-EhPTEN1 enhances cell migration while repression of *EhPTEN1* gene expression causes inhibition of motility. These observations agree well with the fact that EhPTEN1 was transiently concentrated in newly formed pseudopods. The GFP-EhPTEN1 distribution in *E. histolytica* is similar to the localization of mammalian PTEN, which predominantly shows cytosolic localization and mediates conversion of PtdIns(3,4,5)$P_3$ to PtdIns(4,5) $P_2$ through dynamic interaction with the inner face of the plasma membrane [61,62]. In *D. discoideum*, PTEN was implicated in cell migration as a positive regulator of motility, because an ameba strain lacking PTEN showed a reduction in migration speed and defect in chemotactic efficiency due to disruption of PtdIns(3,4,5)$P_3$ / PtdIns(4,5)$P_2$ concentration gradient throughout the cell [63–65]. In contrast, in mammalian cell types including B cells, glioma cells, and fibroblasts, PTEN was shown to inhibit migration [66,8]. Loss of PTEN also resulted in dysregulation of myosin II assembly at the cell cortex, where PTEN prevents the formation of lateral pseudopodia and promotes cell body contraction and posterior retraction in *D. discoideum* [63,67]. In addition, PtdIns(4,5)$P_2$, produced by PTEN, can recruit and activate a wide variety of actin regulatory proteins at the plasma membrane, thereby controlling motility [1,50]. For example, PtdIns(4,5)$P_2$ activates N-WASP directly or indirectly through interaction with IQGAP1 which result in promoting actin polymerization by activation of N-WASP–Arp2/3 complex [1,50,68]. Among them, myosin II and Arp2/3 complex are conserved in *E. histolytica*, where myosin II plays a critical role in movement [60,69] and Arp2/3 complex is involved in actin nucleation [70]. Thus, it is conceivable that EhPTEN1 mediates signaling for pseudopod formation and migration through regulation of PtdIns(3,4,5)$P_3$ metabolism.

In conclusion, we have shown the biological significance of EhPTEN1 in different forms of endocytosis including trogocytosis, phagocytosis, pinocytosis, and clathrin-mediated endocytosis. We have also demonstrated the essentiality of EhPTEN1 in pseudopod formation, motility, and optimal growth of *E. histolytica*. Taken together, these findings emphasize the importance of EhPTEN1 in modulating a plethora of functions in *E. histolytica*. Exploring PTEN functions in *E. histolytica* will hopefully increase our knowledge on the regulation of cellular processes related to actin remodeling through the PIPs signaling pathway.

## Materials and methods

### Identification and comparison of PTEN sequences

Amino acid sequences of PTEN from *E. histolytica* and other organisms were gained from AmoebaDB (http://amoebadb.org/amoeba/) and NCBI (https://www.ncbi.nlm.nih.gov) respectively, and aligned using CLUSTAL W program (http://clustalw.ddbj.nig.ac.jp/) to examine the domain configuration and the key residues for phosphatase activity [71].

## Organisms, cultivation, and reagents

Trophozoites of *E. histolytica* clonal strains HM-1:IMSS cl6 and G3 strain were cultured axenically in 6 ml screw-capped Pyrex glass tubes in Diamond's BI-S-33 (BIS) medium at 35.5˚C as described previously [72–74]. CHO cells were grown at 37˚C in F12 medium (Invitrogen-Gibco, New York, U.S.A.) supplemented with 10% fetal bovine serum on a 10-cm-diameter tissue culture dish (IWAKI, Shizuoka, Japan). *Escherichia coli* BL21 (DE3) strain was purchased from Invitrogen (California, USA). $Ni^{2+}$-NTA His-bind slurry was obtained from Novagen (Darmstadt, Germany). Rhodamine B isothiocyanate-Dextran (RITC-Dextran) and anti-GFP antibody were purchased from Sigma-Aldrich (Missouri, USA). The anti-HA 16B12 monoclonal mouse antibody was purchased from Biolegend (San Diego, USA). Anti-His antibody was purchased from Cell Signaling Technology (Massachusetts, USA). Lipofectamine, PLUS reagent, and geneticin (G418) were purchased from Invitrogen. CellTracker Green, Orange, and Blue were purchased from Thermo Fisher Scientific (Massachusetts, USA). Restriction enzymes and DNA modifying enzymes were purchased from New England Biolabs (Massachustts, USA) unless otherwise mentioned. Luria Bertani (LB) medium was purchased from BD Difco (New Jersey, USA). Other common reagents were from Wako Pure Chemical (Tokyo, Japan), unless otherwise stated.

## Establishment of *E. histolytica* transformants

To construct a plasmid to express EhPTEN1 fused with HA or GFP tag fused at the amino terminus, a DNA fragment corresponding to cDNA encoding EhPTEN1 was amplified by polymerase chain reaction (PCR) from *E. histolytica* cDNA using a pair of primers listed in S2 Table. The PCR-amplified fragments were digested with XmaI and XhoI and cloned into pEhEx-HA and pEhEx-GFP vectors [21,22] that had been predigested with XmaI and XhoI, to produce pEhExHA-EhPTEN1 and pEhExGFP-EhPTEN1. For antisense small RNA-mediated transcriptional silencing of *EhPTEN1* gene, a 420 bp fragment of the protein coding region of *EhPTEN1* gene, corresponding to the amino terminus of the protein, was amplified by PCR from cDNA with sense and antisense oligonucleotides containing StuI and SacI restriction sites (S2 Table). The amplified product was digested with StuI and SacI and ligated into the compatible sites of the double digested psAP2-Gunma plasmid [75] to synthesize a gene silencing plasmid designated as psAP2-EhPTEN1. Two plasmids, pEhExHA-EhPTEN1 and pEhExGFP-EhPTEN1, were introduced into the trophozoites of *E. histolytica* HM-1:IMSS cl6 strain, whereas psAP2-EhPTEN1 was introduced into G3 strain by lipofection as described previously [76]. Transformants were initially selected in the presence of 1 μg/ml G418 until the drug concentration was gradually increased to 10 μg/ml for the *EhPTEN1* gene silenced stain and 20 μg/ml for the GFP- and HA-EhPTEN1 overexpressing stains. Finally, all transformants were maintained at 10 or 20 μg/ml G418 in BIS medium.

## Reverse transcriptase PCR

Reverse transcriptase PCR was performed to check mRNA levels of EhPTEN1 in EhPTEN1 gene silenced and control strains. Total RNA was extracted from trophozoites of EhPTEN1 gene silenced and control strains that were cultivated in the logarithmic phase using TRIZOL reagent (Life Technologies, California, USA). Approximately one μg of DNase treated total RNA was used for cDNA synthesis using Superscript III First -Strand Synthesis System (Thermo Fisher Scientific, Massachusetts, USA) with reverse transcriptase and oligo (dT) primer according to the manufacture's protocol. Ex Taq PCR system was used to amplify DNA from the cDNA template using the primer pairs listed in S2 Table. The PCR conditions were as follow: initial denaturation at 98˚C for 10 sec; then 25 cycles at 98˚C for 10 sec, 55˚C

for 30 sec, and 72˚C for 20 sec; and a final extension at 72˚C for 7 min. The PCR products obtained were resolved by agarose gel electrophoresis.

## Immunoblot analysis

Trophozoites of amoeba transformants expressing HA-EhPTEN1 or GFP-EhPTEN1 grown in the exponential growth phase were harvested and washed three times with phosphate buffer saline (PBS). After resuspension in lysis buffer (50 mM Tris-HCl, pH 7.5, 150 mM NaCl, 0.1% Triton-X 100, 0.5 mg/ml E-64, and protease inhibitor), the trophozoites were kept on ice for 30 min, followed by centrifugation at $500 \times g$ for 5 min. Approximately 20 μg of the total cell lysates were separated on 10% SDS-PAGE and subsequently electrotransferred onto nitrocellulose membranes. The membranes were incubated in 5% non-fat dried milk in Tris-Buffered Saline and Tween-20 (TBST; 50 mM Tris-HCl, pH 8.0, 150 mM NaCl, and 0.05% Tween-20) for 1 hr at room temperature to block non-specific protein. The blots were reacted with one of the following primary antibodies diluted as indicated: anti-HA 16B12 monoclonal mouse antibody at a dilution of 1:1,000, anti-GFP mouse monoclonal antibody (1:100), and anti-CS1 rabbit polyclonal antisera [77] (1:1,000) at 4˚C overnight. The membranes were washed with TBST and further reacted with horseradish peroxidase-conjugated (HRP) anti-mouse or anti-rabbit IgG antisera (1:10,000) at room temperature for 1 hr. After washings with TBST, the specific proteins were visualized with a chemiluminescence HRP Substrate system (Millipore, Massachusetts, USA) using LAS 4000 (Fujifilm Life Science, Cambridge, USA) according to the manufacture's protocol.

## Live cell imaging

Approximately $5 \times 10^5$ trophozoites of the transformant strain expressing GFP-EhPTEN1 were cultured on a 35 mm (in diameter) collagen-coated glass-bottom dish (MatTek Corporation, Massachusetts, USA) in 3 ml of BIS medium under anaerobic conditions. CHO cells were stained with BIS medium containing 10 μM CellTracker Orange for 40 min followed by washing three times with PBS. Approximately $2 \times 10^4$ prestained CHO cells 200 μl BIS were gently overlaid to trophozoites grown on the glass-bottom dish as prepared above. The central part of the dish was then carefully covered with a 1 cm square coverslip and the edge of the coverslip on the slide glass was sealed with nail polish. Live images were captured on Zeiss LSM780 confocal microscope (Carl-Zeiss, Oberkochen, Germany) using a 63× oil immersion objective with default settings on the time series mode and analyzed by ZEN software (Carl-Zeiss). For live imaging of trogocytosis and phagocytosis, approximately twenty cells were analyzed per each cell line in an experiment of three independent experiments. GFP-EhPTEN1 recruitment was estimated by quantitating the average fluorescent intensity per pixel across the line drawn over the cell or that of 11–13 regions of interest on the pseudopods and other parts of the cell.

## Indirect immunofluorescence assay (IFA)

Approximately $5 \times 10^3$ trophozoites in 50 μl BIS were transferred to an 8 mm round well on a slide glass (Matsunami Glass Ind, Osaka, Japan). After 30 min incubation in an anaerobic chamber at 35.5˚C, $5 \times 10^4$ CHO cells that had been pre-stained with 10 μM CellTracker Blue in 50 μl BIS were added to the well and the mixture was incubated for 15 min. After removing the medium, cells were fixed with PBS containing 3.7% paraformaldehyde at room temperature for 10 min, and subsequently permeabilized with PBS containing 0.2% Triton 100-X and 1% bovine serum albumin (BSA) for 10 min each at room temperature. The cells were then reacted with anti-HA mouse monoclonal antibody (1:1000) for 1 hr at room temperature. Then the sample was reacted with Alexa Flouor-488 conjugated anti-mouse IgG (1:1000)

antibody (Thermo Fisher, Massachusetts, USA). The images were then captured using LSM 780 confocal microscope and analyzed by ZEN software (Carl-Zeiss, Oberkochen, Germany).

## Trogocytosis and phagocytosis assay using CQ1

We quantified trogocytosis and phagocytosis efficiency using an image cytometer as described previously [27]. Briefly, trophozoites of *E. histolytica* were incubated in BIS containing 10 μM CellTracker Blue (Thermo Fisher) at 35.5˚C for 1 hr. After staining, ameba trophozoites were washed 3 times with PBS and resuspended in OPTI-MEM medium (Thermo Fisher, Massachusetts, USA) containing 15% adult bovine serum (Sigma Aldrich). Approximately $2 \times 10^4$ ameba trophozoites were seeded into a well on a 96-well glass bottom plate (IWAKI, Shizuoka, Japan) and incubated in anaerobic chamber for 40 min. After incubation, about $1 \times 10^5$ live or heat killed CHO cells that have been stained with 10 μM CellTracker Orange were added to the well containing amebae. The images of trogocytosis and phagocytosis were captured on a Confocal Quantitative image cytometer CQ1 (Yokogawa Electric Corporation, Tokyo, Japan) using a 20× objective every 10 min for 1 hr. Five fields each per well for three wells were chosen for image analysis. Images on five planes with 2 μm intervals in Z axis were obtained to calculate the volume of the ingested CHO cells using three-dimensional reconstructed images. For analysis of motility, images of approximately 400 cells were captured from each well and analyzed using CellPathfinder software (Yokogawa Electric Corporation, Tokyo, Japan) according to the manufacture's protocol. The protocol for quantitative analysis of the images was optimized based on the cell morphology and fluorescence intensity to identify *E. histolytica* trophozoites and CHO cells in each field. The multiple parameters were measured to evaluate the efficiency of trogocytosis and phagocytosis: the average number of internalized CHO cells per amoeba, the combined volume of internalized CHO cells per amoeba, and the percentage of amebic trophozoites that ingested the target cells in the whole population.

## Measurement of fluid-phase and receptor-mediated endocytosis

Approximately $2.5 \times 10^5$ amebic transformants were incubated in BIS medium containing 2 mg/ml fluorescent fluid-phase marker RITC dextran at 35˚C for indicated time points. The cells were collected, washed three times with PBS, and resuspended in 250 μl of lysis buffer (50 mM Tris-HCl, pH 7.5, 150 mM NaCl, and 0.1% Triton-X 100). Fluorescence intensity was measured using a plate reader (SpectraMax Paradigm Multi-Mode, Molecular Devices, California, USA) at an excitation wavelength of 570 nm and an emission wavelength of 610 nm.

Approximately $2 \times 10^4$ amebic transformants were incubated in BIS containing 20 μM CellTracker Blue (Thermo Fisher) at 35.5˚C for 1 hr. After staining, approximately $10^4$ amebic transformants resuspended in 100 μl of BIS medium were transferred to a well on a 96-well glass-bottom plate. After incubation at 35.5˚C in an anaerobic chamber for 40 min, 0.5 mg/ml of transferrin conjugate with Alexa Fluor 568 was added to the well and images were acquired by CQ1 and analyzed as above.

## Migration (motility) assay

Amoebic trophozoites grown in the logarithmic growth phase were harvested and labelled with 20 μM CellTracker Green for 1 hr at 35.5˚C. After washing 3 times with PBS, cells were transferred to a well on a 96-well glass-bottom plate and time lapse images were captured on CQ1. The motility of the cells was measured using CellPathfinder software.

## Quantitative real-time (qRT) PCR

The relative levels of mRNA of *EhPTEN1* gene and RNA polymerase II gene, as an internal standard, were measured by qRT-PCR. PCR reaction was prepared using Fast SYBR Master Mix (Applied Biosystems, California, USA) with 100 ng cDNA and a primer set shown in S2 Table. PCR was conducted using the StepOne Plus Real-Time PCR system (Applied Biosystems, California, USA) with the following cycling conditions: an initial step of denaturation at 95˚C for 20 sec, followed by 40 cycles of denaturation at 95˚C for 3 sec, annealing and extension at 60˚C for 30 sec. The mRNA expression level of *EhPTEN1* gene in the transformants was expressed as relative to that in the control transfected with psAP2.

## Growth assay of *E. histolytica* trophozoites

Approximately $10^4$ trophozoites of *E. histolytica* G3 strain transformed with psAP2-EhPTEN1 and psAP2 (control), grown in the logarithmic phase, were inoculated into 6 ml of fresh BI-S-33 medium containing 10 μg/mL G418, and the parasites were counted every 24 hr on a hemocytometer.

## Production of EhPTEN1 recombinant protein

To construct the plasmid for the production of recombinant EhPTEN1 containing a histidine-tag at the amino terminus, the full-length protein coding sequence of *EhPTEN1* gene was amplified by PCR using oligonucleotide primers listed in S2 Table. PCR was performed with PrimeSTAR Max DNA polymerase (Takara Bio Inc, Shiga, Japan) with the following parameters: initial incubation at 95˚C for 1 min; followed by 30 cycles of denaturation at 98˚C for 10 sec; annealing at 55˚C for 5 sec; and elongation at 72˚C for 15 sec; and a final extension at 72˚C for 30 sec. The PCR fragment was digested with *Bam*HI and *Sal*I and ligated into *Bam*HI and *Sal*I double digested pCold-1 vector (Takara Bio Inc, Shiga, Japan) to produce pCold1-EhPTEN1 plasmid. The pCold-1-EhPTEN1 was introduced into *E. coli* BL21 (DE3) cells by heat shock at 42˚C for 45 sec. *E. coli* BL21 (DE3) strain harboring pCold-1-EhPTEN1 was grown at 37˚C in 50 ml of LB medium (BD Difco, New Jersey, USA) in the presence of 100 μg/ml ampicillin. The overnight culture was used to inoculate 500 ml of fresh medium, and the culture was further continued at 37˚C with shaking at 220 rpm for approximately 2 hr. When $A_{600}$ absorbance reached 0.6, then 1mM of isopropyl β-D-thio galactopyranoside (IPTG) was added, and cultivation was continued for another 24 hr at 15˚C. The *E. coli* cells from the induced culture were harvested by centrifugation at 75,000 rpm for 20 min at 4˚C. The cell pellet was washed three times with PBS, re-suspended in 30 ml of the lysis buffer (50 mM Tris–HCl, pH 8.0, 300 mM NaCl, and 0.1% Triton X-100) containing 100 μg/ml lysozyme, and 1 mM phenylmethyl sulfonyl fluoride (PMSF), and incubated at room temperature for 30 min. After incubation, the mixture was sonicated on ice and centrifuged at 13,000 rpm for 20 min at 4˚C. The supernatant was mixed with 1 ml of 50% $Ni^{2+}$-NTA His-bind resin (Qiagen, Hilden, Germany), incubated for 1 hr at 4˚C with mild rotatory shaking. The resin that recombinant His-EhPTEN1 bound was washed in a disposal column three times with 5 ml of lysis buffer containing 10–30 mM of imidazole. Bound proteins were eluted with 3 ml each of lysis buffer containing 100–300 mM imidazole to obtain recombinant EhPTEN1. The integrity and the purity of the recombinant protein were confirmed with 10% SDS-PAGE analysis, followed by Coomassie Brilliant Blue staining. Then the protein was concentrated, and the buffer was replaced with 50 mM Tris-HCl, 150 mM NaCl, pH 8.0 using Amicon Ultra 50K centrifugal device (Millipore, Massachusetts, USA). The protein was stored at -30˚C with 50% glycerol in small aliquots until further use.

## Lipid phosphatase assay

EhPTEN1 enzymatic activity was determined by the method previously described [78]. Di-C8 phosphatidylinositol phosphate(s) (PIPs) (Echelon Bioscience, Salt Laken City, USA) were dissolved in 100 mM MOPS, pH 6.0, solution, flash frozen in liquid nitrogen, and stored at -20˚C between uses. For determination of pH optimum, the following buffers were used, 100 mM acetate buffer (pH 4.0, pH 4.5, pH 5.0, pH 5,5), 100 mM MOPS buffer (pH 6.0, pH 6.5, pH 7.0), and 100 mM Tris-HCl (pH 7.5, pH 8.0, pH 8.5, pH 9.0). For determination of substrate specificity, a reaction mixture was composed of 25 µl of 100 mM MOPS pH 6.0 containing 5 µg of recombinant EhPTEN1 and 100 µM PIPs. The reaction was carried out at 37˚C for 40 min and the produced phosphate was measured using Malachite Green Reagent (Cell Signaling technology, Massachusetts, USA). After incubation for 15 min at room temperature, the absorbance was measured at a wavelength of 630 nm. The Lineweaver Burk Plot was used to calculate the kinetic parameters of EhPTEN1.

## Lipid membrane overlay assay

Approximately $6\times10^6$ trophozoites of GFP-EhPTEN1 expressing strain were harvested, washed with PBS, and concentrated by centrifugation. Approximately one hundred µl of lysis buffer (50 mM Tris-HCl, pH 7.5 150 mM NaCl, and 0.1% Triton X-100, 1× complete mini, and 0.5 mg/ml E64) was added to the cell pellet. The mixture was incubated on ice for 30 min and centrifuged at 13000 rpm for 5 min. The supernatant was collected and used as the total lysate. GFP-EhPTEN1 was immunoprecipitated using GFP-Trap Agarose Kit (ChromoTek, Planegg, Germany) according to the manufacturer's instruction, eluted, and confirmed by immunoblot. Lipid membranes on which a panel of phospholipids were immobilized (PIP strips: P-6001, Echelon Bioscience, Salt Laken City, USA) were blocked with 1% fat free BSA in PBS-T (PBS containing 0.05% Tween 20) for 1 hr at room temperature. The membranes were then incubated with 2 ml of lipid binding solution (1% fat free BSA in PBS-T, 1x complete mini, 0.05 mg/ml E64, 20 µl of eluted lysate) for 3 hr at 4C. After the membrane were washed twice with PBS-T at 4˚C, they were reacted with anti-GFP at 1:100 dilution with PBS-T containing 1% fat free BSA (PBS-TB) in for 3 hr at 4C. After incubation with the first antibody, the membranes were further reacted with HRP conjugated anti-mouse IgG rabbit antiserum at 1:10,000 dilution with PBS-TB in 4˚C for 1 hr. The membranes were washed three times with PBS-T at 4˚C and the specific proteins were visualized with a chemiluminescence HRP substrate system (Millipore, Massachusetts, USA) using LAS 4000 (Fujifilm Life Science, Cambridge, USA) according to the manufactures' protocol. Recombinant EhPTEN1 protein was also used except that 1 µg/ml of recombinant protein were incubated on the membrane over night at 4˚C and anti-His antibody was used as the first antibody with a dilution of 1:1,000.

## Site directed mutagenesis of EhPTEN1

Overlapping primers were designed to replace cysteine at amino acid 140 of the pEhExGF-P-EhPTEN1 and pCold1-EhPTEN1 plasmids with serine. The primer sets used are listed in S2 Table. PCR reaction was carried out using 0.2 µM/µl each of the primers, 25 µl of PrimeSTAR Max DNA polymerase (Takara Bio Inc, Shiga, Japan) and pEhExGFP-EhPTEN1 or pCold1-EhPTEN1as a template. The PCR parameters were: incubation at 95˚C for 1 min; followed by 30 cycles of denaturation at 98˚C for 10 sec, annealing at 55˚C for 15 sec, and elongation at 72˚C for 50 sec; and a final extension at 72˚C for 30 sec. The PCR fragment was evaluated by agarose gel electrophoresis and introduced into DH5α competent *E. coli* cells by heat shock at 42˚C for 45 sec. Bacterial colonies containing the plasmid that encodes mutated EhPTEN1 (pEhExGFP-EhPTEN1_C140S and pCold-1-EhPTEN1_C140S) were selected from LB plate

containing 100 µg/mL ampicillin by DNA sequencing of the plasmids. The pEhExGFP-EhPTEN1_C140S plasmid was introduced into the trophozoites of *E. histolytica* HM-1:IMSS cl6 strain as described previously [76]. Transformants were initially selected in the presence of 1 µg/ml G418 until the drug concentration was gradually increased to 20 µg/ml. The pCold-1-EhPTEN1_C140S was introduced into *E. coli* BL21 (DE3) cells to produce the inactive mutant recombinant protein as described previously.

## Supporting information

**S1 Fig. Live imaging montage showing localization of GFP-EhPTEN1 in normal motile trophozoites. (A-B)** Montage showing a time series of motile trophozoites expressing GFP-EhPTEN1 in left panels. The pseudopodal localization of GFP-EhPTEN1 is indicated by white arrow. The right panels show the fluorescence intensity of GFP-EhPTEN1 across the amoebic trophozoites. (Scale bar, 10 µm).
(TIF)

**S2 Fig. Live imaging montage showing localization of GFP mock in normal motile trophozoites. (A-C)** Montage showing a time series of motile trophozoites expressing GFP in left panels. The pseudopods in different time frames have been analyzed for GFP intensity along the marked arrow line. The right panels show the fluorescence intensity of GFP across the amoebic trophozoites. (Scale bar, 10 µm).
(TIF)

**S3 Fig. Expression and localization of HA-EhPTEN1. (A)** Immunoblot analysis of HA-EhPTEN1 in *E. histolytica* transformants. Approximately 30 µg of total lysates from mock-transfected control (mock) and HA-EhPTEN1-expressing transformant (HA-EhPTEN1) were subjected to SDS-PAGE and immunoblot analysis using anti-HA antibody. EhCS1 (Cysteine synthase 1) was detected by anti-CS1 antiserum as a loading control. Arrow indicates HA-EhPTEN1. **(B)** Localization of HA-EhPTEN1 in a quiescent state. Immunofluorescence assay (IFA) micrographs of HA-EhPTEN1 expressing trophozoites stained with anti-HA antibody (green). (Scale bar, 5 µm). **(C)** The line intensity plot shows HA-EhPTEN1 intensity in pseudopods vs. cytoplasm with the distance. **(D)** Average fluorescent signal per pixel at different regions in the cells. Fluorescent signal per pixel in approximately 10 circular regions of interest was measured to get the average intensity per pixel. **(E)** Localization of HA-EhPTEN1 during initial phase of phagocytosis. HA-EhPTEN1 expressing *E. histolytica* trophozoites were co-cultured with CellTracker Orange-stained dead CHO cells for 15 min, fixed, and reacted with the anti-HA antibody (green). **(F)** Localization of HA-EhPTEN1 after phagosome maturation. HA-EhPTEN1 expressing *E. histolytica* trophozoites were co-cultured with CellTracker Orange-stained dead CHO cells for 30 min, fixed, and reacted with the anti-HA antibody (green).
(TIF)

**S4 Fig. Localization of GFP mock during phagocytosis. (A)** Montage of live trophozoite expressing GFP ingesting pre-killed CHO cells by phagocytosis. (Scale bar, 10 µm). **(B-D)** Analysis of intensity of GFP across the phagocytic cup along the line drawn.
(TIF)

**S5 Fig. Evaluation of gene expression by RT-PCR analysis of *EhPTEN1* gene silenced transformant.** The steady-state levels of transcripts of *E. histolytica* PTEN isoforms and *EhRNA pol II* genes were measured in psAP2 mock and EhPTEN1gs transformants trophozoites. cDNA from the generated cell lines was subjected to 25 cycles of PCR using specific primers

mentioned in S2 Table. RNA polymerase II served as a control.
(TIF)

**S6 Fig. Expression and purification of EhPTEN1 in *E. coli*. (A)** Expression and purification of recombinant EhPTEN1. Protein samples at each step of purification were subjected to 10% SDS-PAGE and the gel was stained with Coomassie Brilliant Blue. **(B)** Immunoblot analysis of purified recombinant EhPTEN1 using anti-His-tag antibody. The recombinant EhPTEN1 in the supernatant was visualized after longer exposure**. (C)** Optimum pH of EhPTEN1. Enzyme specific activity of recombinant EhPTEN1 was measured at various pHs indicated in the figure. The means ± standard errors of three independent experiments are shown.
(TIF)

**S7 Fig. Lipid binding specificity of EhPTEN1.** Lipid binding specificity of EhPTEN1 observed by lipid overlay assay. A panel of PIPs and phospholipid spotted in nitrocellulose membrane was incubated with total lysates from GFP-EhPTEN1 and GFP mock expressing transformants, and recombinant His-EhPTEN1. LPA, lysophosphatidic acid; LPC, lysophosphocholine; PE phosphatidylethanolamine; PC, phosphatidylcholine; S1P, sphingosine-1-phosphate, PA, phosphatidic acid; PS, phosphatidylserine.
(TIF)

**S8 Fig. Kinetic analysis of EhPTEN1 phosphatase activity. (A)** Saturation Kinetics for EhPTEN1 against $PI(3,4,5)P_3$, $PI(3,5)P_2$, and $PI(3,4)P_2$. Varying amount of $PI(3,4,5)P_3$, $PI(3,5)P_2$, and $PI(3,4)P_2$ were mixed with EhPTEN1 recombinant protein and phosphate release was measured by a malachite green binding assay as mentioned in Materials and methods. **(B)** Double reciprocal plots of the recombinant EhPTEN1. The enzymatic activities were determined with various concentration of $PI(3,4,5)P_3$, $PI(3,5)P_2$, and $PI(3,4)P_2$. Data are shown in means ± standard deviations of duplicate analysis.
(TIF)

**S9 Fig. Expression, localization, and phenotypic analysis of GFP-EhPTEN1_C140S. (A)** Schematic representation of EhPTEN1 and EhPTEN1_C140S mutant. Amino acid residues mutated are shown in red. **(B)** Immunoblot of GFP-EhPTEN1_C140S and GFP (control) in *E. histolytica* transformants. Approximately 30 μg of total lysates from GFP mock-transfected control and GFP-EhPTEN1_C140S expressing transformant were subjected to SDS-PAGE and immunoblot analysis using anti-GFP antibody and anti-CS1 antibody. An arrow indicates GFP-EhPTEN1_C140S. **(C)** Localization of GFP-EhPTEN1_C140S in a quiescent state. (Scale bar, 10 μm). **(D)** The effect of GFP-EhPTEN1_C140S expression on pinocytosis. Trophozoites of GFP mock transfected and GFP-EhPTEN1_C140S expressing strains were mixed with RITC dextran and uptake was monitored as described in Materials and methods. Experiments were conducted twice independently and error bars indicating standard errors of two biological replicates. **(E-F)** Effect of GFP-EhPTEN1_C140S expression on trogocytosis (E) and phagocytosis (F). Trophozoites of GFP mock transfected and GFP-EhPTEN1_C140S expressing strains were incubated with live CHO cells (E) or heat killed CHO cells (F) that have been stained with CellTracker Orange to evaluate trogocytosis and phagocytosis, respectively. The images were captured on CQ1 as described in Materials and methods and analyzed to calculate the numbers of CHO cell-containing trogosomes or phagosomes per amoeba. Experiments were conducted twice independently in duplicates and error bars indicating standard errors of two biological replicates.
(TIF)

**S10 Fig. Production and activity of EhPTEN1_C140S. (A)** Expression and purification of recombinant EhPTEN1_C140S. Protein samples at each step of purification were subjected to 10% SDS-PAGE and the gel was stained with Coomassie Brilliant Blue. **(B)** Immunoblot analysis of purified recombinant EhPTEN1_C140S using anti-His-tag antibody. **(C)** Specific activity of recombinant EhPTEN1 and EhPTEN1_C140S using 100 μM of PtdIns(3,4,5)P3 as substrate. Reactions were carried out in a volume of 25 μl for 40 min at 37˚C, and terminated by the addition of 100 μl of malachite green reagent as described in Materials and methods. The absorbance at 630 nm was measured and phosphate released was quantified by comparison to a standard curve of inorganic phosphate. The mean ± S.D. of three independent experiments is shown.
(TIF)

**S11 Fig. Sequence alignments of PTEN homologs in *Entamoeba histolytica*.** Multiple amino acid sequence alignment of human PTEN (P60484), EhPTEN1 (XP_653141.2), EhPTEN2 (XP_656021.2), EhPTEN3 (XP_655519.1), EhPTEN4 (XP_652082.1), EhPTEN5 (XP_656532.1), and EhPTEN6 (XP_654831.1) was constructed by using clustalw algorithm (http://clustalw.ddbj.nig.ac.jp). PTEN phosphatase domain and C2 domain are shown with blue and yellow backgrounds, respectively. The green rectangle corresponds to the PtdIns(4,5) P$_2$-binding motif (PDM domain). Amino acid residues implicated for PtdIns(3,4,5)P$_3$ catalysis are marked with a red rectangle. Cytosolic localization signal and restudies important for TI loop formation are indicated in black and blue lines, respectively. The DUF457 domain is shown in magenta background, whereas PEST and PDZ-BM are shown in green and cyan background, respectively. Note that for human PTEN, only the amino terminal part is shown.
(TIF)

**S12 Fig. Schematic diagram represents potential function of EhPTEN1 in *E. histolytica* during trogocytosis, phagocytosis, clathrin-mediated endocytosis (CME), micropinocytosis, and migration.** EhPTEN1 localizes mainly in the cytosol with some enrichment in pseudopod-like structures and in the early stage of trogo- and phagocytosis, probably due to changes in the lipid content of the membrane during these events. EhPTEN1 can negatively regulates trogocytosis and phagocytosis by reducing the local PtdIns(3,4,5)P$_3$ concentration leading to the suppression of actin dependent cytoskeletal organization needed for trogo- and phagocytosis. In contrast, EhPTEN1 serves as a positive regulator of fluid-phase and receptor-mediated endocytosis in *E. histolytica* possibly via augmentation of PtdIns(4,5)P$_2$ synthesis which causes recruitment of clathrin regulator and enhances formation of lipid rafts that needed for micropinocytosis.
(TIF)

**S1 Table. Percentage of amino acid identity among *E. histolytica* PTEN isoforms and Human PTEN by ClustalW multiple sequence alignment score.**
(TIF)

**S2 Table. List of primers used in this study. Restriction site marked by bold letter**.
(TIF)

**S1 Movie. Live cell imaging showing localization of GFP-EhPTEN1 in the cytoplasm and pseudopod like structures formed by trophozoites.** The trophozoites expressing GFP-EhPTEN1 proteins were seeded onto 3.5 cm diameter glass bottom dish and then observed using Carl Zeiss LSM780 confocal microscope. (Scale bar, 10 μm)
(MOV)

**S2 Movie. Live cell imaging showing localization of GFP in motile trophozoites.** The trophozoites expressing GFP mock were seeded onto 3.5 cm diameter glass bottom dish and then observed using Carl Zeiss LSM780 confocal microscope. (Scale bar, 10 μm)
(MOV)

**S3 Movie. Live cell imaging showing localization of GFP-EhPTEN1 during trogocytosis.** The trophozoites expressing GFP-EhPTEN1 proteins were co-cultured with CellTracker Orange stained live CHO cells onto 3.5 cm diameter glass bottom dish and then observed using Carl Zeiss LSM780 confocal microscope. (Scale bar, 10 μm).
(MOV)

**S4 Movie. Live cell imaging showing localization of GFP-EhPTEN1 during phagocytosis.** The trophozoites expressing GFP-EhPTEN-1 proteins were co-cultured with CellTracker Orange stained pre-killed CHO cells onto 3.5 cm diameter glass bottom dish and then observed using Carl Zeiss LSM780 confocal microscope. (Scale bar, 10 μm).
(MOV)

**S5 Movie. Live cell imaging showing localization of GFP during phagocytosis.** The trophozoites expressing GFP mock were co-cultured with CellTracker Orange stained pre-killed CHO cells onto 3.5 cm diameter glass bottom dish and then observed using Carl Zeiss LSM780 confocal microscope. (Scale bar, 10 μm).
(MOV)

## Acknowledgments

We thank all members of Nozaki lab, particularly Herbert J. Santos, for technical assistance and valuable discussions, and Arif Nurkanto, Emi Mazaki, Kumiko Shibata, Mihoko Imada, and Ratna Wahyuni for all the help and advice.

## Author Contributions

**Conceptualization:** Kumiko Nakada-Tsukui, Tomoyoshi Nozaki.

**Data curation:** Samia Kadri, Kumiko Nakada-Tsukui.

**Formal analysis:** Samia Kadri, Kumiko Nakada-Tsukui, Natsuki Watanabe, Ghulam Jeelani.

**Funding acquisition:** Kumiko Nakada-Tsukui, Natsuki Watanabe, Tomoyoshi Nozaki.

**Investigation:** Samia Kadri, Kumiko Nakada-Tsukui, Natsuki Watanabe.

**Methodology:** Samia Kadri, Kumiko Nakada-Tsukui, Natsuki Watanabe, Tomoyoshi Nozaki.

**Project administration:** Kumiko Nakada-Tsukui, Ghulam Jeelani, Tomoyoshi Nozaki.

**Resources:** Samia Kadri, Kumiko Nakada-Tsukui, Natsuki Watanabe, Ghulam Jeelani, Tomoyoshi Nozaki.

**Supervision:** Tomoyoshi Nozaki.

**Validation:** Samia Kadri, Kumiko Nakada-Tsukui, Natsuki Watanabe.

**Writing – original draft:** Samia Kadri, Tomoyoshi Nozaki.

**Writing – review & editing:** Tomoyoshi Nozaki.

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
