## [Decision Letter · Decision Letter 0]

19 Dec 2021

Dear Prof. Nozaki,

Thank you very much for submitting your manuscript "PTEN differentially regulates endocytosis, migration, and proliferation in the enteric protozoan parasite Entamoeba histolytica" for consideration at PLOS Pathogens. As with all papers reviewed by the journal, your manuscript was reviewed by members of the editorial board and by several independent reviewers. In light of the reviews (below this email), we would like to invite the resubmission of a significantly-revised version that takes into account the reviewers' comments.

We cannot make any decision about publication until we have seen the revised manuscript and your response to the reviewers' comments. Your revised manuscript is also likely to be sent to reviewers for further evaluation.

Sincerely,

William A. Petri, Jr.

Associate Editor

PLOS Pathogens

Vern Carruthers

Section Editor

PLOS Pathogens

Kasturi Haldar

Editor-in-Chief

PLOS Pathogens

orcid.org/0000-0001-5065-158X

Michael Malim

Editor-in-Chief

PLOS Pathogens

orcid.org/0000-0002-7699-2064

Reviewer's Responses to Questions

**Part I - Summary**

Reviewer #1: The phosphatase and tensin homolog protein (PTEN) acts through a lipid phosphatase activity. PTEN is a phosphatidyl-inositol 3'-kinase (PI 3-kinase) antagonist; the two proteins having important contrasting roles in metabolism and cell growth. PTEN phosphorylate phosphatidyl-inositol (3,4,5) triphosphates [PI (3,4,5) P3) 5, IP (1-3-4-5) 4] and is frequently mutated in most human cancers which lead to increased growth and motility of cancer cells. In this work, a protein from Entamoeba histolytica, showing 39% of identity with PTEN, has been identified and characterized by cell biology and biochemical approaches. EhPTEN1 is enzymatically highly active against PtdIns(3,4,5)P3 and is required for optimal growth of the parasite. EhPTEN1 is involved in the regulation of different modes of endocytosis, namely fluid-phase endocytosis, receptor-mediated endocytosis, phago-, trogocytosis, and cell migration. As such, the data presented in this manuscript are interesting because the essential functions of PTEN in humans are conserved in amoebae, despite the phylogenetic distance between these two organisms. Nevertheless, and according to design of the study, the potential differences between the two proteins are not yet known.

In a general view this work is interesting because the consequences of PTEN on various processes in E. histolytica. It can be noted that beyond the expected activities, in comparison with PTEN in humans, the fact that there is also an effect on important markers of pathogenesis, such as phagocytosis and trogocytosis, is very interesting to advance in the study of these phenomena. The paper is describing interesting data; the experiments are conducted with appropriated controls and modern methodologies

Reviewer #2: In this study, a PTEN homolog (one of six candidates in the genome) was studied in Entamoeba histolytica. Localization was assessed using a GFP-tagged protein in migrating trophozoites, and in trophozoites undergoing trogocytosis or phagocytosis. Expression of the GFP-tagged protein led to increased cell motility, decreased trogocytosis, decreased phagocytosis, and increased pinocytosis. Conversely, gene silencing led to decreased cell motility, increased trogocytosis, increased phagocytosis, decreased pinocytosis, and decreased growth rate. Phosphatase activity of the recombinant protein was demonstrated, and dot blots were used to assess binding of the overexpressed tagged-protein from trophozoite lysates to various lipid substrates.

This study expands understanding of phospholipid signaling in E. histolytica. Many of the phenotypes assessed in this study are consistent with phenotypes associated with PTEN activity in other organisms, making the results not necessarily surprising, but nonetheless important to establish in this poorly understood human pathogen.

Reviewer #3: PTEN is involved in lipid signalling through manipulation of levels of phosphoinositol phosphates. There are seven different forms of PIPs and their levels are intricately maintained through a number of kinases and phosphatases. PTEN is a phosphatase. It has been known for quite some time that different PIPs play important role in amoebic physiology, particularly during different endocytic mechanisms including phagocytosis and trogocytosis. In this manuscript, the authors have attempted to show the involvement of one of the E. histolytica PTENs in phagocytosis and trogocytosis. Delineation of these two pathways is highly desirable as this may help to find new drug targets as well as help us understand the biology of E. histolytica. Though the evidence in support of the conclusion comes mainly from phenotypic characterization after PTEN downregulation and overexpression, these do not conclusively show that the observed effects are due to the enzyme activity of PTEN.

**Part II – Major Issues: Key Experiments Required for Acceptance**

Reviewer #1: 1. The first experiment concerns the choice of the version of the isoform of PTEN which is studied according to the levels of transcription of genes corresponding to the 6 isoforms determined by bioinformatics. It has been done taking advantage of previous work from the team during description of the transcriptome of E. histolytica using microarrays technology. The question arises as to how, among the oligonucleotides that cover the gene sequence; the chip discriminates among the six isoforms of the transcript? To reinforce the assumption of high gene expression level for the chosen PTEN, it should be of interest to inform readers about the key sequence and/or region within each transcript that discriminates the six transcripts. I did a search using AmoebaDB resources (https://amoebadb.org) with the six gene ID and the results from multiple RNASeq experiments are: EHI_197010 is the more expressed gene, which corroborates the authors conclusion, but expression of EHI_041900 and EHI_098450 genes is also high. A better explanation of the transcription rate should be presented. The further question in this part is what is the rationale to look at PTEN expression in the G3 silenced strain? There is not comment on this finding in the main text.

2. In humans, PTEN is a protein of 403 amino acids that consists of five essential domains.

a-The PIP-binding domain that localizes at the N-terminal

b-The phosphatase domain containing the HCXXGXXR motif, which is the catalytic sequence in protein tyrosine phosphatases.

c-The N-terminal first 190 amino acids are homologous to the actin-binding protein tensin

d-The C2 domain that bind phospholipids in membrane independent of calcium

e-The C-terminal domain consists of 2 PEST sequences, including phosphorylated serine-threonine spots and a PDZ domain

According to Figure 1, domains a, b and d are present in EhPTEN1 and are well described in the manuscript. Unfortunately, there is no information on the tensin part inserted in the C2 domain and as well as, the consequences of the absence of the PDZ domain. In the biology of PTEN, it has been described that a truncated protein lacking the C-terminal PDZ binding motif, or proteins which extend beyond the PDZ binding motif, have been modified in the function of PTEN, including during cancer metastases (PMID: 10760291). The domains of tensin and PDZ deserve attention in this context.

3. A further look at the other potential PTEN indicate that the nomenclature proposed to distinguish them in E. histolytica proteins can lead to confusion. The absence of C2 domains indicates that two of these potential proteins lack lipid binding activity. Are these bona fide PTEN? In humans, variations of PTEN derive from the same loci (PMID: 24768297, PMID: 23744781) after the start of alternative translation. In E. histolytica, isoforms 1 and 2 appear to be homologous (48.8% identity in Table S1). I have run the CLUSTALW program with the mRNA sequences of both genes and there is 64% identity between them; moreover, in isoform 2 the DNA sequence presents introns. This is important because in this work it is shown that E. histolytica retained at least 4 isoforms of PTEN, many more than in the other species studied. Perhaps a more restrictive nomenclature should help avoid future confusion. Additionally, more in-depth analysis of other PTEN isoforms could be provided by at least benefiting from transcriptome analysis, along with proteomic analyzes from the team and other groups, to convince readers that these proteins exist.

4. Regarding cell biology experiments like live cell imaging and image analysis, several shortcomings deserve the author's attention: It is mandatory to describe the experimental setting for live cell imaging, microscopy approaches and software implementations. For example, which confocal plane, which path for image analysis, what is CQ1? What references and what algorithms are used in the CellPathfinder software. As like other technical approaches, the reader needs references for compounds, companies, and step-by-step methods. May I respectfully tell the authors to look at the article PMID: 32111929 in the topic PTEN, which, in my opinion, describes the imaging and image analysis approaches very well.

5. I understand the difficulties of studying gene expression and protein abundance in E. histolytica due to the lack of genetic tools to obtain stable phenotypes. Here, the overexpression of PTEN is obtained by the insertion of two hybrid genes: GFP-PTEN (useful for monitoring GFP in live cell imaging) and HA-PTEN (useful for monitoring PTEN in immunofluorescence of fixed using HA antibodies). These two constructs were introduced into the strain E. histolytica HM-1: IMSS c16.

Enrichment of GFP-EhPTEN1 in the pseudopod of living trophozoites has been observed. Their shifts in speed of displacement were 0.54 ± 0.09 µm / sec (mean ± S.D.), which was significantly greater than that of transformants expressing control GFP (0.27 ± 0.08 µm / sec) ~roughly 2 times.

The effect of repressing the expression of the EhPTEN1 gene was obtained by inserting a silencing construct into the G3 strain, which was shown, at the start of work (FIG. 1B), expressing the gene encoding PTEN twice. In strain G3, transformed by the silencing construct targeting EhPTEN1, the displacement speed is reduced (0.16±0.07 μm /min) compared to the mock control (0.44±0.08 μm /min) ~roughly 2,7 times. If we compare the two mock controls: GFP (0.27 ± 0.08 µm / sec) and G3 (0.44±0.08 μm /min) the change is 1.6.

I could be wrong, but how do you consider these discrete changes as important in the regulation of cell motility? In addition, the strains have different basal level of expression of the PTEN gene. For these reasons, the conclusion regarding cell motility seems weak. Inhibition of PTEN in Dictyostelium alters cell morphology (PMID: 29602807); as well as the nature of their displacements. Further analysis of video microscopies of PTEN-deficient cells, to examine the morphology and direction of trajectories, may help to strengthen this data. Moreover, in some cases, the inhibition of Pi3K can "rescue", at least in part, the phenotype deficient in PTEN.

6. Immunofluorescence on fixed trophozoites overexpressing HA-EhPTEN1 (using anti-HA antibodies) led the authors to say that HA-EhPTEN1 was localized in the cytoplasm at steady state and enriched in pseudopodia (Fig S3B and S3C). Here, the calculations were carried out along a line crossing the cell, but the choice of this line is biased because it appears that similar fluorescence levels are also observed in the back part of the cells. The statement is misleading. A better approach should be to determine fluorescence (pixels) in several regions of interest inside the cells and then compared these values. The speed of movement of these cells has not been determined.

7. My last comment refers to the fact that PTEN is a PI 3-kinase antagonist; the two enzymes having important contrasting roles in metabolism and cell growth. Additionally, some articles examine the role of Pi3K in E. histolytica during phagocytosis and human cell death (eg, PMID: 25246714, PMID: 9317033). I am surprised that the Pi3K antagonistic role of PTEN was not considered in this work. At least this question could be raised in the discussion chapter.

Reviewer #2: Many experimental details that are critical for rigor and reproducibility are missing (mainly related to the number of cells/images analyzed and the number of biological repeats performed). These details are necessary in order to be able to assess the robustness of the conclusions of these studies. There are four main points to address: (1) It is critical to add information on how many cells were analyzed for the imaging experiments and how representative the examples are of the population level phenotypes. Many of the conclusions appear to be based on the analysis of a single example cell shown in the figures (for example, the localization of the tagged proteins during trogocytosis and phagocytosis in Fig 3 and Fig 4). (2) Heterogeneity in tagged protein expression levels is also evident in the movies and still images (meaning there is one bright cell that was analyzed, and many dimly expressing cells in the surrounding imaging field), but this heterogeneity is not addressed or accounted for. (3) In many of the experiments for which statistical analysis was performed (Fig 2d, 5, 6, 7d, 8, 9, 10, 11), the explanation of what is plotted is vague. In these cases, the number of biological repeats that were performed is stated, but it is not clearly explained whether the figures reflect one repeat (i.e., the error bars reflect technical replicates), or the average of all of the biological repeats (i.e., the error bars reflect biological replicates. (4) It is not clear how many cells were analyzed for the CQI analysis or how cells were selected for quantitative analysis.

The localization data for the EhPTEN1 are not strong enough to support the authors’ conclusions about where these proteins localize during trogocytosis and phagocytosis. The imaging quality/resolution is very low, and it also appears that in some cases, what appears to be dynamic localization of the proteins over time is actually just a random change in the plane that was imaged. Higher resolution images from immunofluorescence analysis capturing many cells at various stages of trogocytosis and phagocytosis would be much more conclusive, particularly if combined with co-localization analysis of actin or other relevant proteins. These types of studies would be the strongest if the data clearly reflect assessment of many individual cells.

Reviewer #3: 1. PTEN is one of the enzymes that is involved in PI3P levels and is part of the system that maintains an intricate balance of all PIPs. It is, therefore, important to include a summary of our understanding of different PIPs and the proteins that participate in E. histolytica. The authors should indicate the effect of increasing or decreasing the amount of PTEN to levels of other PIPs and experimentally determine the levels of these metabolites after overexpressing and down-regulating the enzyme.

2. In order to demonstrate that the enzymic activity of PTEN is responsible for its’ phenotype, it is important to overexpress inactive enzyme (mutated) and then observe the phenotype of the cells.

3. It is not clear what are the downstream pathways finally regulate endocytic processes. Questions such as the mechanism of recruitment of PTEN in pseudopods should be addressed.

4. There should be an inactive mutant control for the enzyme assays (Table 1).

**Part III – Minor Issues: Editorial and Data Presentation Modifications**

Reviewer #1: NA

Reviewer #2: For figure 1B, please explain more clearly in the figure legend what the assay is.

The authors used “mock” to refer to control cells throughout the manuscript. More information is needed on what these mock cells are. Are they WT HM1:IMSS trophozoites or G3 trophozoites? Have they been transfected with the corresponding empty vectors or not?

The phosphatase activity of the recombinant protein is the highest for a different phospholipid than those phospholipids that are bound by the overexpressed protein from trophozoite lysates. Please comment on the differences and potential biological meaning of the differences.

The argument that these studies could lead to the development of new control measures for Entamoeba histolytica seems a stretch. If the authors would like to make this argument, this would need to be strengthened with a stronger justification.

Reviewer #3: Fig. 1: The multiple alignment has been done with only three sequences that display fairly large diversity. It should be redone with a number of sequences from different species and all PTEN homologs should be included.

Discussion: It can be reduced and a working model to be included.

PLOS authors have the option to publish the peer review history of their article (what does this mean?). If published, this will include your full peer review and any attached files.

Reviewer #1: No

Reviewer #2: No

Reviewer #3: No
---

## [Editor Report · Decision Letter 1]

4 Apr 2022

Dear Prof. Nozaki,

We are pleased to inform you that your manuscript 'PTEN differentially regulates endocytosis, migration, and proliferation in the enteric protozoan parasite Entamoeba histolytica' has been provisionally accepted for publication in PLOS Pathogens.

Best regards,

William A. Petri, Jr.

Associate Editor

PLOS Pathogens

Vern Carruthers

Section Editor

PLOS Pathogens

Kasturi Haldar

Editor-in-Chief

PLOS Pathogens

orcid.org/0000-0001-5065-158X

Michael Malim

Editor-in-Chief

PLOS Pathogens

orcid.org/0000-0002-7699-2064
---

## [Editor Report · Acceptance letter]

26 Apr 2022

Dear Prof. Nozaki,

We are delighted to inform you that your manuscript, "PTEN differentially regulates endocytosis, migration, and proliferation in the enteric protozoan parasite Entamoeba histolytica," has been formally accepted for publication in PLOS Pathogens.

Best regards,

Kasturi Haldar

Editor-in-Chief

PLOS Pathogens

orcid.org/0000-0001-5065-158X

Michael Malim

Editor-in-Chief

PLOS Pathogens

orcid.org/0000-0002-7699-2064